# Widespread perturbation of ETS factor binding sites in cancer

Sebastian Carrasco Pro [1,7], Heather Hook[2,7], David Bray[1], Daniel Berenzy[3], Devlin Moyer [1], Meimei Yin[2], Adam Thomas Labadorf[4,5], Ryan Tewhey [3], Trevor Siggers [1,2,6,8] ✉ & Juan Ignacio Fuxman Bass [1,2,8] ✉

Although >90% of somatic mutations reside in non-coding regions, few have been reported as cancer drivers. To predict driver non-coding variants (NCVs), we present a transcription factor (TF)-aware burden test based on a model of coherent TF function in promoters. We apply this test to NCVs from the Pan-Cancer Analysis of Whole Genomes cohort and predict 2555 driver NCVs in the promoters of 813 genes across 20 cancer types. These genes are enriched in cancer-related gene ontologies, essential genes, and genes associated with cancer prognosis. We find that 765 candidate driver NCVs alter transcriptional activity, 510 lead to differential binding of TF-cofactor regulatory complexes, and that they primarily impact the binding of ETS factors. Finally, we show that different NCVs within a promoter often affect transcriptional activity through shared mechanisms. Our integrated computational and experimental approach shows that cancer NCVs are widespread and that ETS factors are commonly disrupted.

Cancer initiation and progression are often associated with environmentally induced or spontaneous mutations, and inherited genomic variants that increase cancer risk[1–3]. Large scale projects such as the Cancer Genome Atlas (TCGA) and the International Genome Consortium (ICGC) have identified millions of somatic variants in tumors[4–6]. However, in most cases, it is not known whether these mutations affect any cellular function, confer growth advantage, or are causally implicated in cancer development[7]. The difficulty in annotating variants is that only a few cancer driver mutations are needed to initiate tumor growth, development, and metastasis and these mutations must be distinguished from thousands of passenger mutations that do not alter fitness[7]. Even though more than 90% of somatic variants are in non-coding regions, few non-coding cancer drivers have been identified[6,8,9], highlighting the need for approaches to identify and validate non-coding variants (NCVs) in cancer.

Mutational burden tests have been used to predict driver NCVs. These tests are based on determining an increased mutational frequency in DNA regions of interest (e.g., cis-regulatory elements (CREs)) compared to a background mutational frequency[10–18]. Methods have employed a range of different parameters to estimate the background mutational frequency in CREs, including cancer-specific mutational signatures, sequence conservation, functional annotations, mutational frequencies in neighboring regions or other "similar" genomic regions, replication timing, and expression levels[9,19]. Despite these varied approaches to estimate mutational burden and the increasing number of sequenced tumor samples, studies have only identified ~100 driver NCVs. For example, burden tests within specific cancer types have identified NCVs in the promoters of *TERT*, *FOXA1*, *HES1*, *SDHD*, and *PLEKHS1*[20–22]. Further, a global analysis of 2568 cancer whole genome samples from the Pan-Cancer Analysis of Whole Genomes (PCAWG) identified driver NCVs in the promoters of *TERT*, *HES1* and seven additional genes[9]. A more recent analysis of 3949 tumors from PCAWG and the Hartwig Medical Foundation identified driver NCVs in the promoters and enhancers of 52 genes[19]. Additionally, driver NCVs have been identified in the super-enhancers of *BCL6*, *BCL2*, *CXCR4* in diffuse large B-cell lymphomas[23]. Whether this somewhat limited number of

[1]Bioinformatics Program, Boston University, Boston, MA, USA. [2]Department of Biology, Boston University, Boston, MA, USA. [3]The Jackson Laboratory, Bar Harbor, ME, USA. [4]Bioinformatics Hub, Boston University, Boston, MA, USA. [5]Boston University School of Medicine, Department of Neurology, Boston, MA, USA. [6]Biological Design Center, Boston University, Boston, MA, USA. [7]These authors contributed equally: Sebastian Carrasco Pro, Heather Hook. [8]These authors jointly supervised this work: Trevor Siggers, Juan Ignacio Fuxman Bass. ✉e-mail: tsiggers@bu.edu; fuxman@bu.edu

driver NCVs is due to a modest contribution of NCVs to cancer or to limitations of current approaches to identify and validate NCV drivers remains to be determined.

NCVs in CREs likely affect the binding of transcription factors (TFs) and the recruitment of regulatory cofactors (COFs) leading to changes in gene expression[8]. For example, *TERT* overexpression, a major contributor to cancer, is caused by multiple NCVs in its promoter that create ETS factor binding sites[24–27]. We hypothesize that an approach to assess NCV burden that accounts for changes in TF binding may improve the sensitivity to detect mutational burden.

Here, we present a TF-aware burden test (TFA-BT) based on the assumption that creating (or disrupting) binding sites for a particular TF at different positions within a CRE will have similar transcriptional effects and should therefore be grouped together in the burden analysis. Indeed, it has been reported that TF binding sites (TFBSs) in CREs frequently occur in homotypic clusters and regulate gene expression through cooperative and non-cooperative mechanisms[28,29]. We applied our TFA-BT to promoter NCVs from the PCAWG datasets and predicted 2555 cancer driver NCVs in the promoters of 813 genes across 20 cancer types. These genes are enriched in cancer-related and essential genes, and their expression levels are associated with cancer prognosis. To evaluate our TFA-BT NCVs, we used an integrative approach that combines two high-throughput experimental approaches to assay the impact of NCVs on gene expression and the disruption of TF-COF regulatory complexes. Using MPRAs (massively parallel reporter assays) we found that 765 TFA-BT NCVs altered transcriptional activity, which is a similar validation rate to known driver NCVs. Further, using the microarray-based CASCADE (comprehensive assessment of complex assembly at DNA elements) assay, we found that 510 TFA-BT NCVs lead to differential binding of TF-COF regulatory complexes, and impact primarily the binding of ETS factors. Together, our integrated computational and experimental approach shows that cancer NCVs are a more widespread driver mechanism than previously recognized.

## Results

### Prediction of cancer driver NCVs

We developed a TFA-BT that identifies CREs containing a higher-than-expected number of NCVs across patients that alter (i.e., create or disrupt) TFBSs for a particular TF. We applied our TFA-BT to somatic NCVs in the promoters of protein-coding genes (from −2000 to +250 bp of the transcription start site). Briefly, for each TF-promoter pair (A, B) our method counts the number of NCVs predicted to alter the binding of a specific TF (A) within a promoter (B). We then determine the probability of this observation given (1) the total number of observed NCVs in promoter B across a set of patient samples, and (2) the probability that a random NCV in B (according to the mutational frequency in the patient samples) alters a binding site for TF A (Fig. 1a). These TF-promoter pair probabilities are then used to calculate corrected p-values to identify increased mutational burden in particular promoters. Three different threshold-based approaches were used to predict differential TF binding to NCV alleles and for the subsequent steps in the TF-ABT. For robustness we only considered as TFA-BT NCVs those that were deemed significant (FDR < 0.01) using all three approaches (see Methods). We note that in TFA-BT the mutational burden in the promoter itself, rather than other similar or neighboring genomic regions, functions as background to determine enrichment for altered TF binding. This reduces the need to identify and model the appropriate confounding factors into the burden test, and results in increased power to identify potential driver NCVs.

We applied the TFA-BT to predict cancer driver NCVs (hereafter referred to as TFA-BT NCVs) in the promoters of protein-coding genes using 2654 tumor samples from the PCAWG cohort corresponding to 20 cancer types[6]. Predictions were performed per cancer type and in a pan-cancer analysis. In total, we predicted 2555 TFA-BT NCVs in the promoters of 813 genes, which altered the binding sites of 404 TFs

(Supplementary Data 1). Most TFA-BT NCVs (65%) were obtained from skin cancer (Fig. 1b). This is not only related to skin cancer samples having the largest number of promoter NCVs, but also to a higher fraction of these being predicted as TFA-BT NCVs (Supplementary Fig. 1a). The majority of TFA-BT NCVs (76%) are associated with the disruption, rather than gain, of TFBSs. This is likely related to the disruption of a TFBS having a higher likelihood of being functional and selected in cancers, as we have previously observed that random gain and loss of TFBSs in CREs have similar likelihoods[30].

We observed a wide range of TFA-BT NCVs per gene (Fig. 1c). In some cases, such as the highly mutated *BCL2* and *BCL6*, individual TFA-BT NCVs are generally not recurrent but affect the binding of the same TFs at different positions in the promoter across tumor samples. In other cases, such as *TERT*, a few TFA-BT NCVs are highly recurrent including the widely reported chr5:1295228 C > T and chr5:1295250 C > T mutations (Fig. 1c, see insert)[24,27]. We detected TFA-BT NCVs in multiple genes with reported driver NCVs in promoters, including the highly mutated *PLEKHS1*, *CDC20*, *DPH3*, and *BCL6*[19,21,23,31,32] (Supplementary Fig. 1b). We also found genes that, to our knowledge, have no previously reported driver NCVs with TFA-BT NCVs in at least 5% of tumors within certain cancer types, such as *RPL13A* (bladder and skin cancer), *TEDC2* (skin cancer), and *PES1* (skin cancer) (Supplementary Fig. 1b).

Multiple lines of evidence showed that our TFA-BT gene set is associated with known cancer-related genes, pathways, and functions. First, we detected a significant enrichment in cellular fitness genes[33], essential genes[34], and genes whose expression has been associated with favorable or unfavorable cancer prognosis[35], which was overall higher than for the well-curated lists of Cancer Gene Census and IntOGen genes (Fig. 1d)[36,37]. Second, we identified a significant overlap with genes whose somatic copy number variation is associated with changes in their expression across multiple cancer types (OR = 1.42, $p = 0.007$)[38]. Finally, we found a significant enrichment in gene ontologies associated with general and cancer-related cellular processes (Supplementary Fig. 1c). Interestingly, although many gene ontology terms overlap between TFA-BT and IntOGen genes (a set of genes with driver coding mutations), multiple terms are more enriched in either gene set (Fig. 1e). For example, terms associated with translation and rRNA processing are more enriched within TFA-BT genes, whereas cell cycle, signaling, and transcription terms are more enriched in IntOGen genes. This suggests that non-coding and coding mutations may affect genes with different functions.

### TFA-BT NCVs alter transcriptional activity

To determine whether the TFA-BT NCVs affect transcriptional activity, we evaluated the 2555 TFA-BT NCVs and control NCVs using massively parallel reporter assays (MPRAs)[39,40] in Jurkat (lymphoma), SK-MEL-28 (melanoma), and HT-29 (colorectal) cell lines, which match the cancer types with the most TFA-BT NCVs (Fig. 2a). NCVs that had statistically significant allelic skew between the reference and alternate alleles were called expression-modulating variants (emVars)[41] (Supplementary Data 2). Since only a subset of DNA regions are active (show MPRA activity for either allele – 1378 for Jurkat, 1118 for SK-MEL-28, and 1144 for HT-29 cells), we calculated the validation rate as the ratio of emVars over the total number of active DNA regions for each NCV category. For the TFA-BT NCVs, we detected emVars for 53%, 27%, and 33% NCVs (q < 0.05) for Jurkat, SK-ML-28, and HT-29 cells, respectively, which highly overlap between cell lines (Fig. 2b and Supplementary Fig. 2a). This validation rate is higher than for NCVs with no predicted differential TF binding (Fig. 2b 'No differential binding') or random NCVs with predicted differential TF binding (Fig. 2b 'Non-driver differential TF binding'). The high validation rates for the TFA-BT NCVs are similar to experimentally reported driver NCVs in promoters (Fig. 2b 'Reported driver NCVs'), NCVs leading to an allelic imbalance in ChIP-seq experiments (Fig. 2b 'ChIP-seq allelic imbalance'), and disease-associated germline NCVs that lead to altered target gene expression

and cause differential TF binding (Fig, 2b 'Reported germline NCVs'). Altogether, these results show that the TFA-BT can prioritize functional NCVs.

Most burden tests can identify genomic regions enriched in cancer mutations but cannot determine which of the many mutations in a particular region are actually functional. Interestingly, TFA-BT NCVs validated at a higher rate than random patient-derived NCVs in the promoters of genes reported to have high mutational burden (Fig. 2b 'Random NCVs in reported genes'), suggesting that TFA-BT can better pinpoint functional NCVs. TFA-BT can also be used to predict likely functional NCVs. We tested the transcriptional activity of random NCVs that correspond to significant TF-promoter pairs by TFA-BT but that were not observed in the PCAWG cohort (Fig. 2b 'TFA-BT - unobserved'). These unobserved NCVs validated at a higher rate than random NCVs in reported genes, suggesting that TFA-BT also has predictive value for NCVs not observed in the cohort of study.

Recurrency is often used as a criterion to prioritize cancer mutations. Interestingly, we found that the validation rate for TFA-BT NCVs is similar regardless of the NCV frequency across cancer samples (Fig. 2c). This suggests that NCVs with low mutation frequency, such as those private to particular tumor samples, can also lead to altered transcriptional activity. The power of TFA-BT to predict functional

private mutations is important given that most cancer mutations are private as well as most TFA-BT NCVs (Fig. 2d).

We validated TFA-BT NCVs associated with both the gain and loss of TFBSs. However, we observed a higher validation rate for NCVs that lose TFBSs (56%, 35%, and 29% in Jurkat, HT-29, and SK-MEL-28 cells, respectively) than for NCVs that gain TFBSs (40%, 21%, and 14% in Jurkat, HT-29, and SK-MEL-28 cells, respectively) or NCVs that lead to gain and loss of TFBSs (46%, 24%, and 23% in Jurkat, HT-29, and SK-MEL-28 cells, respectively) (Supplementary Fig. 2b). This difference may be related to a higher likelihood of affecting expression by disrupting an existing TFBS in a CRE than by creating a TFBS that may not be in the appropriate CRE context or distance/orientation to other TFBSs to affect transcriptional activity.

Most driver NCVs have been identified and characterized in core promoter regions (−250bp to +250 bp from the TSS)[9,21]. Here, we used extended promoter regions of −2kb to +250 bp from the TSS, expanding the current analysis landscape. Although the fraction of NCVs in PCAWG is mostly homogenous throughout the extended promoter region, we observed an enrichment of TFA-BT NCVs in the core promoter, even though our model did not incorporate any additional information beyond TF specificities and promoter sequence (Supplementary Fig. 2c). This suggests that considering core promoter

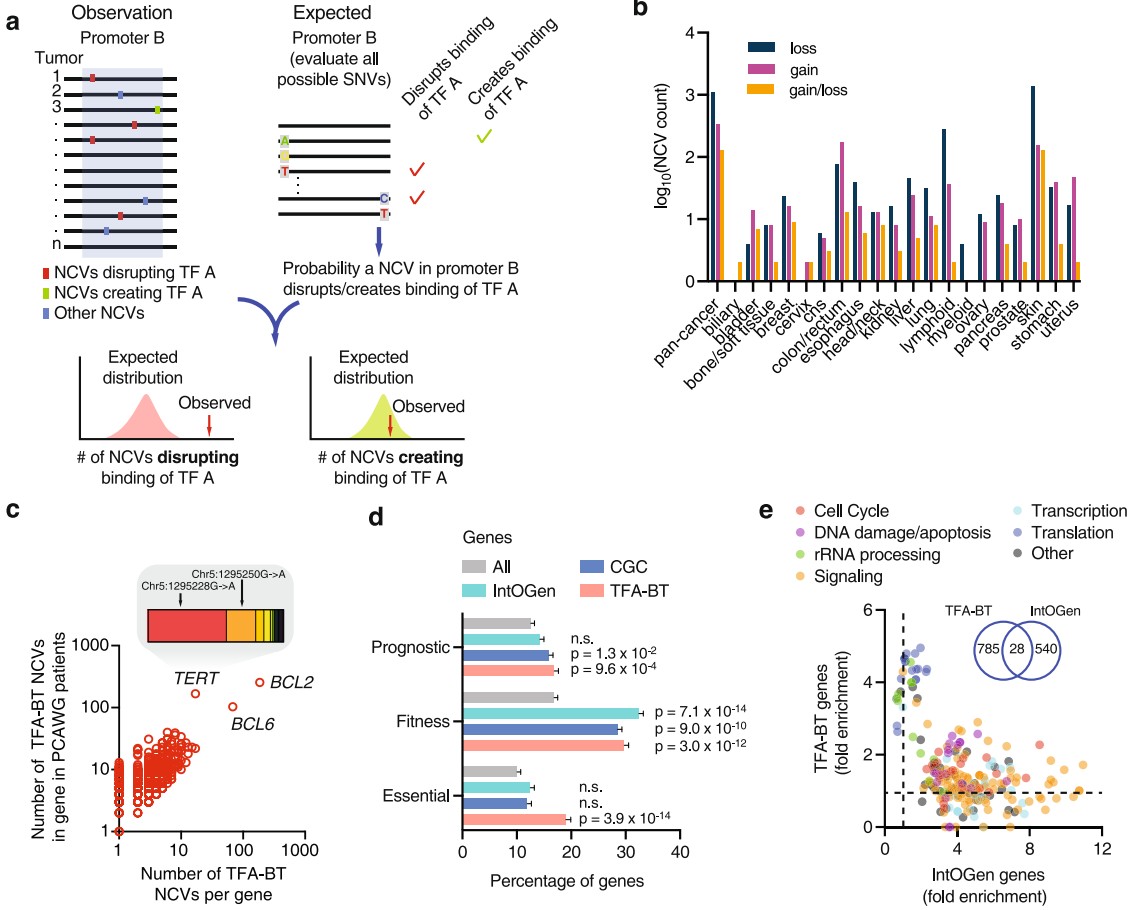

**Fig. 1 | Identification of TFA-BT NCVs. a** Overview of the TFA-BT approach. The number of observed NCVs across tumor samples that disrupt (or create) a binding site of TF A in promoter B is compared to the expected probability distribution to identify significant promoter-TF associations. **b** Number of TFA-BT NCVs with predicted gain and/or loss of TF binding per cancer type. **c** Scatter plot showing the number of different TFA-BT NCVs per gene in the PCAWG cohort versus the number of TFA-BT NCV events in the corresponding promoter in patients from PCAWG. Insert shows fraction of patients in PCAWG for each mutation in the *TERT* promoter. **d** Percentage of prognostic (i.e., genes whose expression levels are

favorably or unfavorably associated with cancer), fitness-related, and essential genes within all protein-coding ($n = 19,208$), IntOGen ($n = 561$), Cancer Gene Census (CGC, $n = 729$), and TFA-BT genes ($n = 746$). Statistical significance determined by two-sided Fisher's exact test compared to all protein-coding genes. Error bars indicate standard error of the proportion. **e** Biological process gene ontology fold enrichment associated with different terms for IntOGen and TFA-BT gene sets. Each dot represents a gene ontology term classified into general classes. Insert shows overlap between TFA-BT and IntoGen genes. Source data are provided as a Source Data file.

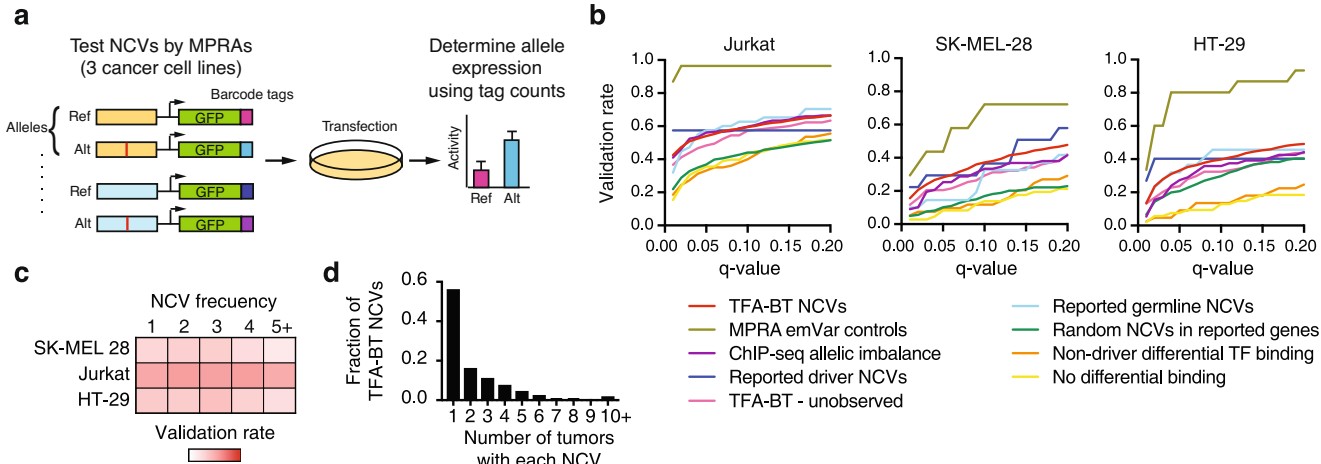

**Fig. 2 | TBA-BT NCVs alter transcriptional activity. a** Overview of the evaluation of NCVs by massively parallel reporter assays (MPRAs). **b** Fraction of NCVs from each test set within MPRA active regions that show expression allelic skew at different q-value thresholds in Jurkat, SK-MEL-28, and HT-29 cells. **c** Heatmap of validation rates in each cell line for NCVs present in 1, 2, 3, 4, and 5 or more patients. **d** Fraction of TFA-BT NCVs per recurrency (i.e., number of tumors with each NCV) across patients in PCAWG. Source data are provided as a Source Data file.

regions likely identifies most driver NCVs in gene promoters. Nevertheless, 25.8% of detected MPRA-validated TFA-BT NCVs reside outside the core promoter (upstream of −250 from TSS), suggesting that interrogating sequences beyond core promoters can also identify functional NCVs.

To determine whether our TFA-BT NCVs affect gene expression in patient tumor samples, we compared target gene expression between PCAWG tumor samples with and without TFA-BT NCVs in their promoters. We limited our analysis to genes with TFA-BT mutations in at least five tumor samples with available expression data within a particular cancer type (Supplementary Fig. 3a). We found 11 genes, including *BCL2*, *DERL1*, and *IFI44L*, in which the presence of TFA-BT NCVs was associated with differences in gene expression (FDR < 0.05, Supplementary Fig. 3a). Notably, we did not observe increased *TERT* gene expression associated with previously reported driver NCVs. This is likely because the analyzes are complicated by the difficulty in comparing gene expression levels across biological samples, the small number of samples with available TFA-BT NCVs per gene, and the stage in cancer development when altered gene expression may occur (e.g., during initial stages in cancer development). To determine whether changes in gene expression caused by TFA-BT NCVs were allele-specific, we measured expression allelic imbalance. We focused on *BCL2* as this was the gene with the highest number of patients with TFA-BT NCVs in PCAWG and found that lymphoma patients with TFA-BT NCVs in the *BCL2* promoter showed a marked allelic expression imbalance of *BCL2* not observed in patients with other or no NCV in the BCL2 promoter (Supplementary Fig. 3b). This supports the gene expression data results and our general conclusion that TFA-BT NCVs are predictive of changes in endogenous gene expression.

**Profiling the impact of NCVs on gene regulatory complexes**
A primary mechanism by which NCVs alter gene expression is by altering the binding of TF-COF regulatory complexes. To examine the mechanism of our TFA-BT NCVs, we profiled their ability to alter the binding of TF-COF complexes. To do this, we employed the recently described CASCADE method in which protein-binding microarrays (PBMs) incubated with cell nuclear extracts are used to profile the differential recruitment of regulatory COFs (e.g., BRD4) to Ref/Alt DNA probe sets[42] (Fig. 3a and Supplementary Fig. 4). As COFs interact broadly with many TFs[43–45], profiling a single COF can report on many DNA-bound TF-COF complexes in a parallel manner without requiring knowledge of the TFs involved. The CASCADE approach provides a

mechanistic annotation to our TFA-BT NCVs that can be integrated with functional MPRA annotations.

To identify differentially bound NCVs, we profiled the recruitment of six COFs spanning a range of functional categories: SRC1 (NCOA1) is a transcriptional coactivator with acetyl-transferase activity; BRD4 is a chromatin reader and regulatory scaffold; MOF (KAT8) is a histone acetyltransferase; NCOR1 is a transcriptional corepressor; RBBP5 is a core member of the MLL/SET histone methyltransferase complexes; TBL1XR1 is a member of the NCoR corepressor complex. COF recruitment was profiled using nuclear extracts from Jurkat and SK-MEL-28 cells to 2956 paired Ref/Alt probe sets that included: 2555 TFA-BT NCVs, 17 literature-reported driver NCVs, and 384 background NCVs predicted to not impact TF binding. NCVs that lead to significant differential recruitment (either gain or loss) of any single COF were classified as a bmVar (binding-modulating variant) (Fig. 3b, Supplementary Fig. 5, Supplementary Data 3).

Of the 2956 assayed NCVs, we identified 513 bmVars: 510 TFA-BT NCVs, two literature-annotated driver NCVs, and one background NCV (Fig. 3c). Critically, bmVars were differentially enriched across the three allele probe groups (Pearson Chi-square test: $p < 7.18 \times 10^{-20}$), with highest bmVar enrichment in our predicted TFA-BT group which was enriched well beyond the background NCVs. Our CASCADE approach is cell-type dependent, and results will vary based on the expression levels and interaction strengths of the TFs and COFs assayed. We identified more bmVars using Jurkat cell extracts but the general trends across probe groups were consistent for both cell types. Of the 510 TFA-BT bmVars we identified, the majority were disruptions in which the NCV led to loss of binding (Fig. 3d). We found that many bmVars were supported by profiles from multiple COFs (Fig. 3e), suggesting that either the disrupted TF is interacting with multiple COFs or multiple TF-COF complexes are disrupted by the NCV. To determine whether our differential TF-COF binding data may explain observed gene expression differences, we determined the overlap between our 510 bmVars and 765 emVars identified for the 2555 TFA-BT NCVs assayed by MPRAs and CASCADE (Fig. 3f). We found 47.0% (359 / 765) of the emVars were also characterized as bmVars in CASCADE, despite only six COFs being profiled. This highly significant overlap ($p$-value = $4.3 \times 10^{-102}$ by hypergeometric test, 2.4-fold-enriched) demonstrates that alteration of regulatory complex binding is strongly predictive of a change in gene expression (i.e., 70%; 359 / 510) and suggests possible mechanisms for the observed gene expression effects. Importantly, TFA-BT genes with NCVs classified as emVars or

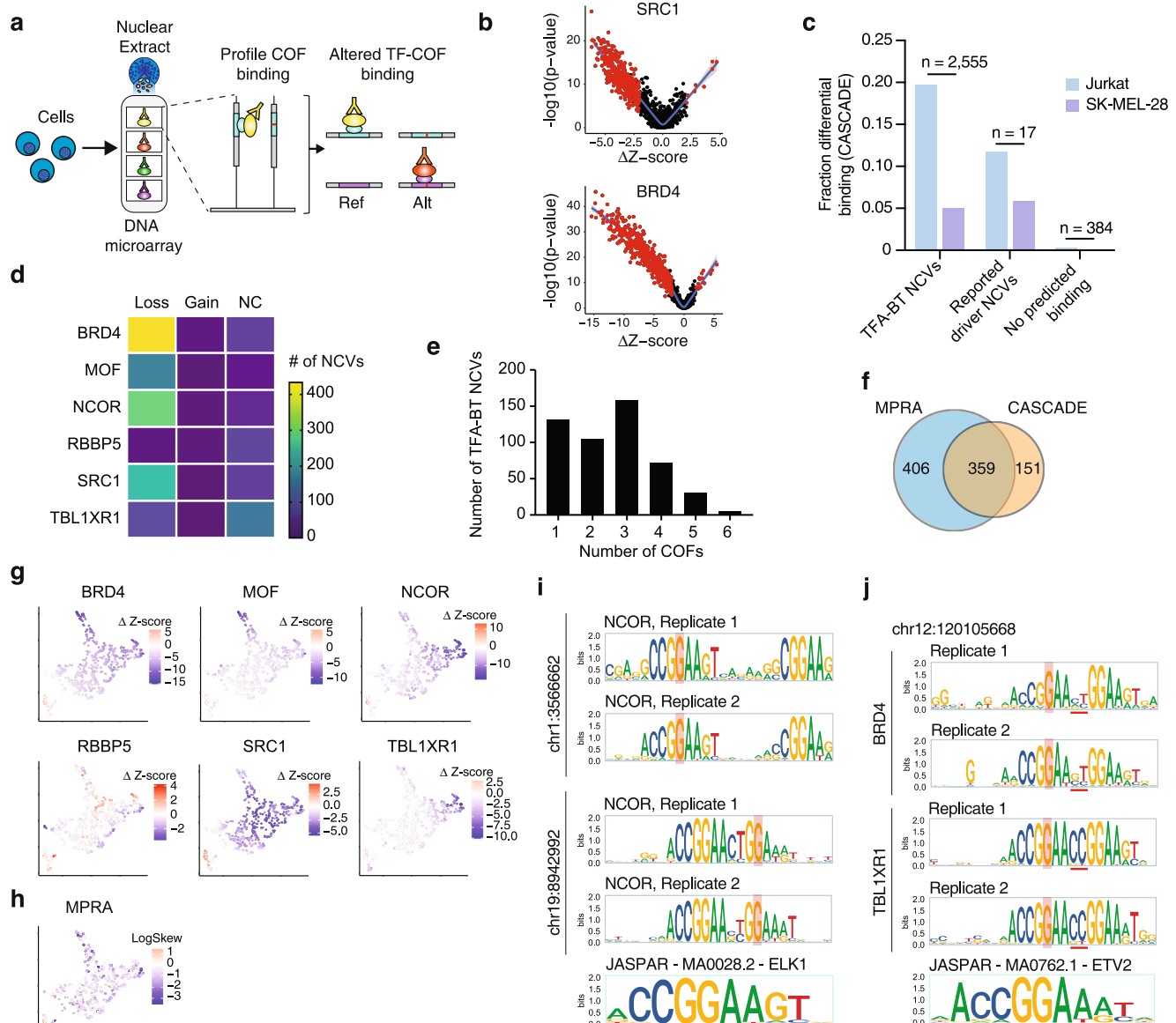

**Fig. 3 | Profiling TF-COF complex binding altered by NCVs. a** Overview of the CASCADE method to profile TF-COF complex binding affected by NCVs (Ref - reference and Alt - alternative alleles). **b** Impact of TFA-BT NCVs on the recruitment of SRC1 and BRD4 to 2555 Ref/Alt NCV probes sets assayed using Jurkat T-cell nuclear extracts. Impact is quantified using -$\log_{10}$(p-value) of the COF recruitment to the different probe sets and the difference in PBM-determined Z-score between Ref and Alt alleles (Δz-score). P values are calculated using two-sided Student's t-test comparing five replicates of Ref and Alt alleles. The NCVs identified as significant are highlighted in red. **c** Fraction of NCVs from different probe sets identified as significant by CASCADE in Jurkat and SK-MEL-28 cells. Numbers at the top of the bars indicate the number of probes tested in each set. **d** Number of TF-ABT NCVs leading to loss, gain, or no change (NC) (i.e., both alleles similarly recruit the COF) of recruitment for each COF tested. **e** Number of TFA-BT NCVs that affect the recruitment of 1 to 6 COFs. **f** Overlap between the number of TFA-BT NCVs significant by MPRAs and CASCADE. **g** UMAP clustering TFA-BT NCVs based on Δz-score for each of the six COFs tested. **h** UMAP depicting the MPRA expression allelic skew for each TFA-BT NCV. **i** NCOR recruitment motifs associated with two TFA-BT NCVs. **j** BRD4 and TBL1XR1 recruitment motifs associated with NCV at position chr12:120105668. Source data are provided as a Source Data file.

bmVars displayed a higher enrichment in essential, fitness, and prognostic genes than all TFA-BT genes (Supplementary Fig. 6). This suggests that these functional NCVs impact genes with important roles in cell viability and cancer.

To examine the relationships between COF dependence and gene expression we used UMAP to represent NCVs based on their impact on COF binding (Fig. 3g). This functional representation of NCVs highlights that NCVs vary in their influence on the recruitment of different COFs. For example, MOF and TBL1XR1 are most strongly disrupted by different sets of NCVs. Mapping the NCV impact on gene expression (i.e., logSkew values from MPRA analysis) onto this COF-binding

representation, we find relatively uniform distribution throughout, suggesting that gene expression data as measured by a reporter assay is not strongly correlated with the impact on a particular COF (Fig. 3h). This data suggests that transcription can be impacted by altering the binding of complexes with diverse COF recruitment characteristics.

## TF-ABT NCVs primarily affect the binding of ETS factors
Our TFA-BT approach is based on identifying NCVs that alter TF binding motifs. In our original analysis, we predicted TFBS alterations for 404 TFs from multiple TF families. For 48.7% of the NCVs we predicted binding changes in two or more TFs, and for some NCVs up to

62 TFs. Therefore, prediction alone is not sufficient to determine the TF whose binding is altered by an NCV. To address the identity of the TF affected by each NCVs, we used CASCADE to determine binding motifs impacted by the 359 NCVs identified as significant by both CASCADE and MPRAs (Fig. 3f, Supplementary Data 4). To do this, we assayed COF recruitment to all single-nucleotide variants spanning each NCV loci and determined recruitment motifs that can be used to infer the underlying TFs by matching against TF motif databases (Supplementary Fig. 7)[42]. We profiled recruitment of our six COFs, using Jurkat nuclear extracts, and determined COF recruitment motifs for 273 loci (Methods). 98% of the COF motifs matched ETS-family motifs, while the remaining ones resembled ETS motifs but matched similar looking motifs (e.g., IRF and STAT family motifs).

Most of the identified motifs are single ETS motifs with the NCV disrupting this single binding site (Supplementary Fig. 8). However, we also identified 18 composite ETS sites where two motifs occur together or separated by up to seven bases (i.e., GGAA-N-GGAA, N = 2,3,5,6,8,9) (Fig. 3i, j). The presence of composite ETS sites is consistent with their tendency to cluster in human promoters[46]. Motifs were consistent across COF experiments (Fig. 3i, j and Supplementary Fig. 8), demonstrating that the different COFs are recruited by either the same ETS protein or by different ETS proteins to the same site(s). While motifs agree well across COFs, we did find evidence of COF-specific base preferences at some loci. In the *PARS2* promoter, for two sites, we found that BRD4 was recruited to an extended ETS motif with additional 5-prime-flank base preferences compared to NCOR (Supplementary Fig. 8). Another example is seen for a composite ETS site where we found that TBL1XR1 and BRD4 differed in their preferences for the 2-bp spacer between the sites, with TBL1XR1 preferring the canonical CC bases while BRD4 preferences were more degenerate (Fig. 3j). These COF-specific preferences provide a mechanism for the differential impact of NCVs on COF recruitment at the same loci and highlight the complexity of determining mechanisms for individual NCVs even for the same class of TFBSs.

## NCVs derived from highly prevalent mutational processes affect transcriptional activity and COF recruitment

Somatic mutations are caused by endogenous and exogenous mutational processes that differ between patients and cancer types leading to different mutational signatures[1,47]. We examined the possible mutational processes generating our TFA-BT NCVs using the PCAWG mutational signature assignments. 58% of TFA-BT NCVs were associated with the SBS 7a, 7b, 7c, and 65 UV-light mutational signatures, consistent with most NCVs being identified in skin cancer (Fig. 4a). We also found 7.4% of NCVs were associated with POLE signatures (SBS61, SBS62, and SBS10a frequently present in colorectal cancers) and 1.4% were associated with APOBEC signatures (SBS2 and SBS13). These highly prevalent signatures, which frequently lead to hypermutation, are often filtered either prior to the burden test or post-test to determine driver NCV candidates[9,21]. Interestingly, we found that NCVs associated with many of these signatures (SBS 7a, 7b, 7c, 13, 61, and 65) validate by MPRAs at similar or higher rates than other TBA-BT NCVs (Fig. 4a). This suggests that many NCVs excluded from other burden test analyzes are potentially functional, affecting transcriptional activity and COF recruitment (Fig. 4a). In particular, NCVs associated with UV-light mutational signatures validate at a higher rate than NCVs not associated with UV-light (Fig. 4a, b). These UV-light TFA-BT NCVs are enriched at the GG doublet in the 5′-GGAA-3′ consensus site and downstream flanking sequence, as previously reported (Fig. 4c)[48,49]. However, their effect on gene expression and COF binding has not been fully addressed. We found that these frequently mutated bases, in particular the two Gs in the 5′-GGAA-3′ consensus ETS site, also correspond to the positions with the largest perturbation in transcriptional activity and COF binding (Fig. 4c). Although this is generally consistent across COFs, we found that mutations in the second G rarely disrupt and often increase

RBBP5 binding. This suggests that the binding of different COFs may be differentially perturbed at different positions of the ETS motif. Further, we found that position information content does not necessarily correlate with functional changes, as mutations in the first A in the 5′-GGAA-3′ consensus site rarely perturb transcriptional activity and COF binding (Fig. 4c). Altogether, this shows a complex interplay between mutations, transcriptional activity, and COF binding and underscores the need for extensive COF profiling.

## Mechanistic similarities and differences between NCVs within promoters

Multiple TFA-BT NCVs in a gene promoter often led to similar transcriptional effects (over or under expression). For example, all validated NCVs in the *TERT* promoter led to increased transcriptional activity, consistent with previously characterized *TERT* promoter drivers associated with *TERT* overexpression[24,27] (Supplementary Fig. 9). Conversely, all validated TFA-BT NCVs in the *EGR1* and *RNF20* promoters led to reduced transcriptional activity (Fig. 5a, b and Supplementary Fig. 9). This is consistent with under expression of *EGR1* and *RNF20* being reported in multiple cancer types[50–52]. For example, *RNF20* under expression due to promoter hypermethylation has been previously associated with genome instability in multiple cancer types[50,53,54]. Our results suggest that reduced *RNF20* promoter activity resulting from NCVs constitutes another potential cancer mechanism.

Similar changes in transcriptional activity between NCVs within a promoter can either be related to similar changes in COF recruitment or to different COF recruitment patterns. We found that NCVs within a promoter have a more similar effect on COF recruitment patterns than NCVs between promoters (Fig. 5c). For example, four of five NCVs in the *EGR1* promoter led to reduced recruitment of BRD4, MOF, NCOR, and SCR1, showing mechanistic convergence between different mutations within the same promoter (Fig. 5a). This convergence can, in some cases, be explained by NCVs being in close proximity (<10 bp), likely affecting the same TFBS; however, other NCVs that similarly alter COF recruitment are located tens of bp away (Fig. 5a, d chr5:137800743 and chr5:137800840, and Fig. 5b, e chr9:104296044 and chr9:104296134). Although there is an overall similarity in altered COF recruitment between NCVs in a promoter, we also observed multiple cases where NCVs in a promoter alter the recruitment of overlapping but different sets of COFs (Fig. 5a, b and Supplementary Fig. 8). This suggests that either a few overlapping COFs may be primarily responsible for the observed transcriptional effect or that different COFs can lead to similar transcriptional effects. Finally, we detected NCVs with altered transcriptional activity where none of the COFs tested showed altered recruitment (Fig. 5a and Supplementary Fig. 8b). We hypothesize that these NCVs may affect transcriptional activity through altered recruitment of other COFs not profiled in our assay.

## Discussion

In this study, we developed a TFA-BT which we applied to 2654 tumor samples from the PCAWG cohort[6] and predicted 2555 driver candidates in the promoters of 813 genes. This is 10- to 20-fold more NCVs and genes than what has been previously reported[9,19–22], showing the power of our TFA-BT approach. Importantly, one third of the TFA-BT NCVs displayed expression allelic skew in MPRAs, a similar rate to well-characterized somatic driver and germline NCVs. Further, this is likely a conservative estimate given that our MPRAs (i) only evaluate a small 200 bp sequence fragment and are missing neighboring chromatin context[39,41], (ii) many (40%) NCVs reside in elements that do not exhibit activity by MPRA and are thus unable to be evaluated, and (iii) we evaluated only three cell lines in this study. We also found that one fifth of the TFA-BT NCVs lead to altered DNA binding of TF-COF complexes assayed by CASCADE. This is also likely a conservative estimate as only six COFs were profiled and NCVs show COF specificity. Altogether, these results show that the TFA-BT can prioritize NCVs that lead to

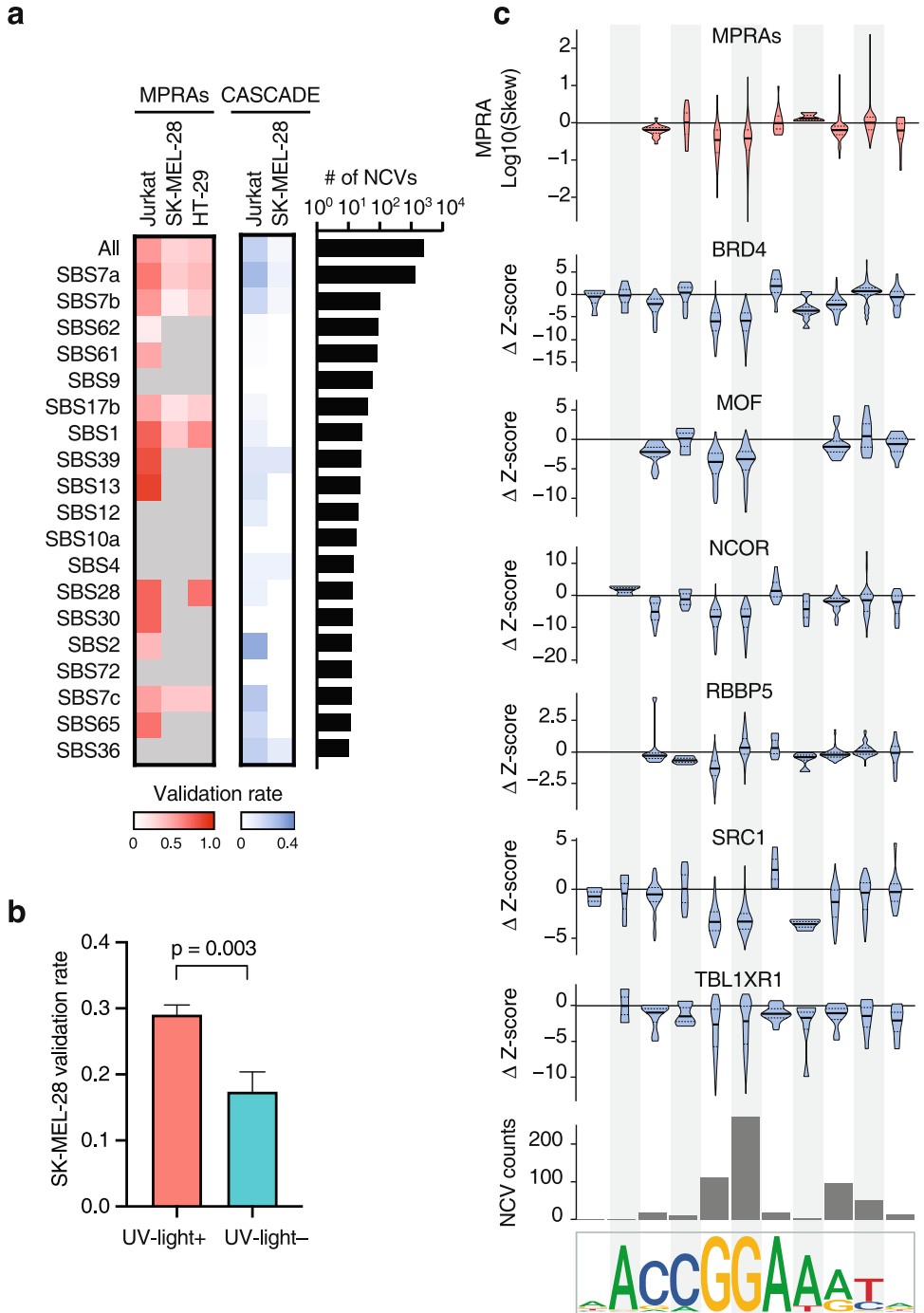

**Fig. 4 | NCVs derived from highly prevalent mutational processes affect transcriptional activity and COF recruitment. a** MPRA and CASCADE validation rates for TFA-BT NCVs associated with different mutational signatures. Only mutational signatures associated with five or more NCVs in MPRA active regions in at least one cell line are shown. Gray cells indicate mutational signatures with less than 5 NCVs in MPRA active regions in the indicated cell line. The right heatmap depicts the fraction of TFA-BT NCVs in each mutation signature that are associated with altered COF recruitment. **b** MPRA validation rate for NCVs associated or not with ultraviolet (UV)-light mutational signature in SK-MEL-28 cells. UV-light + NCVs ($n = 967$), UV-light – NCVs ($n = 161$). Error bars indicate the standard error of a proportion.

Significance determined by two-sided Fisher's exact test. **c** Mutational frequency and effect on transcriptional activity and COF binding for skin cancer TFA-BT NCVs depending on the position within the ETS motif. The top violin plot shows the $log_{10}$ expression allelic skew by MPRA for NCVs affecting different positions within ETS motifs. The bottom six violin plots show the $\Delta z$-score in COF binding between the reference and the alternative allele based on the position of the NCV within the ETS motif. The median is indicated by the bold horizontal line, and the first and third quartiles are indicated by the dotted horizontal lines. The bar plot indicates the number of TFA-BT NCVs affecting each position in the ETS motif. Source data are provided as a Source Data file.

altered gene expression and binding of regulatory complexes. The success of the TFA-BT approach highlights the importance of using regulatory models in NCV burden tests.

Genes containing TFA-BT NCVs are enriched in translation and rRNA processing genes. Mutations in the promoters of these genes may alter their expression leading not only to changes in protein synthesis which can affect cell proliferation, but also to an imbalance in ribosome components and free ribosomal proteins. Free ribosomal proteins caused by altered gene expression or copy number variation have been shown to affect cell cycle, apoptosis, and DNA repair leading

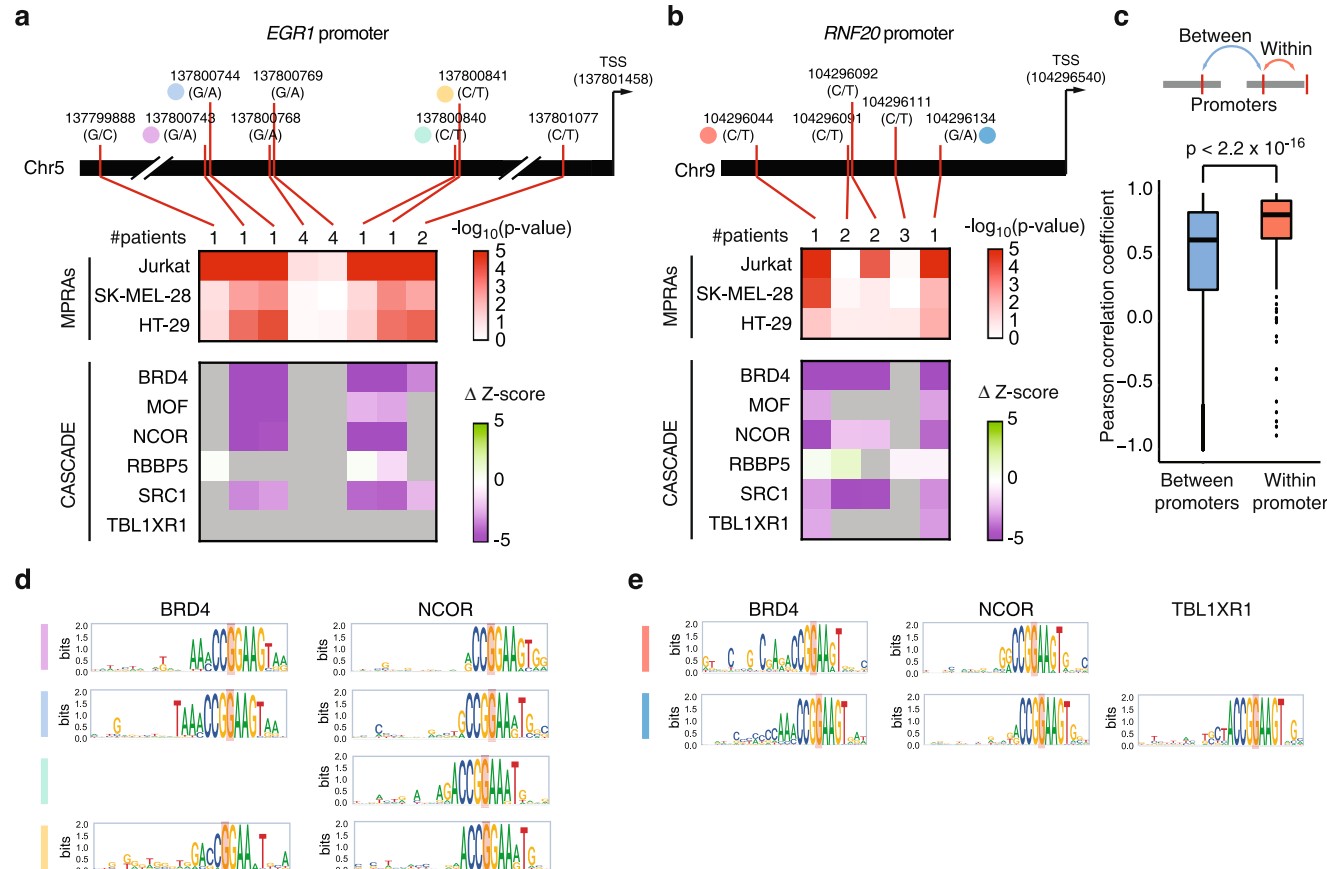

**Fig. 5 | Altered transcriptional activity and COF recruitment within promoters. a, b** Changes in MPRA activity and COF recruitment for TF-ABT NCV in the (**a**) *EGR1* and (**b**) *RNF20* promoters. The top heatmaps show the $\log_{10}(p\text{-value})$ of expression allelic skew in MPRA in Jurkat, SK-MEL-28, and HT-29 cells. *P* values were calculated using two-sided Student's t-test. The bottom heatmaps show the altered COF recruitment by CASCADE, which is indicated as Δz-score. Gray cells indicate cases where the COF was not recruited to either NCV allele. Numbers at the top of the heatmaps indicate the number of patients in PCAWG carrying the indicated NCV. Mutation and TSS coordinates are indicated. **c** Pearson correlation coefficient

(PCC) between Δz-score in CASCADE for each COF between pairs of TF-ABT NCVs within a gene promoter ($n = 510$) and between gene promoters ($n = 258,810$). Each box spans from the first to the third quartile, the horizontal lines inside the boxes indicate the median valuen, the whiskers indicate 1.5x the interquartile range, and the points indicate outliers. Significance determined by two-sided Mann–Whitney U test. **d**, **e** COF recruitment motifs determined by single nucleotide variant scanning using CASCADE for the NCVs indicated in a-b. Source data are provided as a Source Data file.

to cancer[55–57]. Our results suggest that mutations in the promoters of translation genes constitute a potential cancer mechanism.

Most of the TFA-BT NCVs for which we detected altered transcriptional activity reduced gene expression in MPRAs. Given that the vast majority of cancer mutations are heterozygous, this suggests that a partial reduction in the expression of most TFA-BT genes may be sufficient to have a functional role in cancer. Indeed, the haploinsufficiency of multiple genes caused by copy number variation or promoter methylation has been widely associated with cancer[58,59]. Interestingly, we found that 52 of the TFA-BT NCVs are biallelic (49-fold enrichment versus biallelic mutations in PCAWG)[60] and 290 pairs of TFA-BT NCVs are within 10 nt and affect the same TFBS in at least one donor. This suggests that in many cases, TFA-BT NCVs affect both alleles either at the same nucleotide position or at different positions within a TFBS, likely leading to biallelic disruption of gene expression.

We found that NCVs impacting gene expression and regulatory complex binding primarily disrupted ETS-factor binding sites. This is consistent with the known role of ETS factors in cancer initiation and progression[61–63]. Increased and decreased activity of different ETS factors has been implicated in all stages of tumorigenesis via diverse mechanisms, including gene rearrangement and amplification, feed-forward signaling loops, gain-of-function co-regulatory complexes, and cis-acting NCVs in ETS target gene promoters[64]. Our studies further identified the disruption of ETS binding sites in gene promoters as

a widespread cancer mechanism. Whether disruption of ETS factor binding sites is also frequent in enhancers, which often bind a different TF repertoire than promoters, remains to be determined. A large fraction of ETS binding disruption is associated with UV-light mutational signatures and are concentrated primarily in the GG doublet of the canonical 5′-GGAA-3′ ETS box and downstream bases, as has been reported[48,49]. Mutations at these positions have been associated with increased mutational rates at sites of ETS factor binding and potentially reduced DNA repair[65,66], but are mostly considered non-functional and are, therefore, excluded from most burden tests. Here, we show that these frequent ETS-disrupting mutations have the largest transcriptional effects and disruption of COF binding. This suggests that excluding these mutations, as well as those associated with other mutational signatures such as APOBEC and POLE, may not be warranted.

TFA-BT is based on the hypothesis that creating (or disrupting) a TFBS at different positions within a gene promoter is likely to lead to similar effects on target gene expression. However, some of these NCVs may reside in TFBSs that are not bound or functional in vivo. We consider this not to be the major driver of our findings as non-functional NCVs would, in general, not be enriched across patients given that TFA-BT considers the overall promoter mutational burden as background. Another possibility is that binding sites predicted to affect the same TF in a promoter may actually bind TF paralogs with

different effector functions. However, this does not seem to occur frequently, as most TFA-BT NCVs in a promoter tend to perturb transcriptional activity in the same direction (activation or repression).

Although TFA-BT is focused on individual TFs, NCVs that affect the binding of different TFs within a promoter can also have a similar effect on gene expression. This may be the case for NCVs within a promoter that alter the recruitment of similar COFs. Indeed, we found that different TFA-BT NCVs within a promoter often share similar changes in COF recruitment, suggesting shared mechanisms. This supports a potential extension of our approach to develop a COF-aware burden test. This type of test would require knowledge of the COFs that are highly active in a tumor sample as well as the TFs involved in the recruitment of such COFs. Future studies incorporating information on TF-COF complexes will allow us to extend our predictions to other CREs and TFs that may not necessarily function through homotypic clusters.

## Methods

### Altered transcription factor binding predictions
To predict the effect of all possible NCVs in the human genome on TF binding, for each possible NCV and each TF with available position weight matrices (PWMs), we determined the binding score corresponding to the reference and alternative sequences. We downloaded 1898 PWMs corresponding to human TFs from CIS-BP on April 3, 2018[67] and their corresponding TF family. Given a PWM of length $n$ and a genomic position (hs37d5 from the 1000 Genome Project), for each of the $2n-1$ DNA sequences on each strand of length $n$ that overlap with the genomic position, we determined a TF binding score using the function:

$$F(s,M) = \sum_{i=1}^{n} \log\left(\frac{M_{s_i,i}}{b_{s_i}}\right) \quad (1)$$

where $s$ is a genomic sequence of length $n$, $M$ is the PWM with $n$ columns and each column in $M$ contains the frequency of each nucleotide in each position i = 1,…,$n$, and $b_{si}$ is the background frequency of nucleotide $si$ assuming a uniform distribution. The highest score obtained for the $4n$-$2$ sequences ($2n$-$1$ sequences in forward and reverse strands) was assigned as the binding score corresponding to the PWM for the reference or alternate NCV alleles. Significant scores were selected and reported based on TFM-pvalue[68] score thresholds determined using a significance level $\alpha = 10^{-4}$. This method was applied for each reference position and the three possible alternative alleles for the entire human genome (hs37d5) to create an altered TFBS database, a genome-wide catalog of NCV-TF effects. Custom C scripts were developed to generate this dataset using GPUs and the data was stored in the Hadoop servers at Boston University (www.github.com/fuxmanlab/altered_TFBS).

### ChIP-seq allelic imbalance analysis
To estimate optimal threshold(s) of motif scores differences for a given PWM between a reference allele and alternative allele to predict allelic imbalance in TF binding, we used available ChIP-seq experimental data. ChIP-seq experiment FASTQ files were downloaded from the ENCODE Project[69] for 14 datasets (55 experiments) performed in cell lines with normal karyotype (Supplementary Data 5). The files were aligned using BWA[70] and pre-processed using standard GATK methodology[71]. Variant calling was performed on the aligned BAM files using GATK Variant Discovery pipeline (v2)[71] and BCF Tools (v.1.9)[12]. The intersection of variants from both tools was used to extract the allele read counts for each variant. Allelic imbalance analysis was performed for heterozygous positions in promoters for each experiment. A binomial test was used to identify NCVs located in positions where reads were not evenly distributed (0.5 for each allele).

Differential predicted binding events were calculated by comparing the motif score of each alternative to its reference allele. Thresholds of two types were generated for gain/disruption of TFBSs to determine

their ability to predict ChIP-seq allelic imbalance: 1) when only the reference or alternate allele pass the binding threshold for the motif determined by TFM-pvalue[68], or 2) when at least one allele passed the motif binding threshold and the difference in score between alleles (allele score) is above a certain value ranging from 0 to 7. To benchmark our predictions, for each TF, we used NCVs in allelic imbalance in ChIP-seq as true positives and those not in allelic imbalance as true negatives, and compared to predicted gain/loss of TFBSs in the same direction as the allelic imbalance. F-values and relative accuracies were calculated for all thresholds. Based on the F-values, we selected three parameter settings: 1) either the reference or alternate allele pass the binding threshold for the motif determined by TFM-pvalue, 2) at least one allele passed the motif binding threshold and the difference in score between alleles was greater than two, and 3) at least one allele passed the motif binding threshold and the difference in score between alleles was greater than three. These three parameter settings were independently used for the TF-aware burden test (TFA-BT) as described below in section titled "Development of the TF-aware burden test."

### Processing of PCAWG mutational data
We downloaded VCF files of 2654 samples from the PCAWG cohort[6] using the ICGC portal[5] (Jan 23, 2019). We considered only single-nucleotide variants (SNVs) and excluded multi-nucleotide variants from the analyzes. To identify NCVs in promoter regions, we used BEDTools (v.2.27.0) intersection command[72]. Promoters from protein-coding genes were defined as regions between −2 kb to +250 bp from the transcription start sites (TSSs) annotated in GENCODE v19[73]. In the case of overlapping alternative promoters, promoter regions were merged to prevent over-counting. To avoid considering protein-coding regions, in the case of alternative promoters, we filtered "coding_regions" using the GENCODE v19[73] (Jun 14, 2018) annotation. We used the R package IRanges (v.2.12)[74] to determine the promoter coordinates, and BEDTools (v.2.27.0)[72] was used to remove promoter coordinates overlapping with coding regions (Supplementary Data 6).

### Development of the TF-aware burden test
We designed the TFA-BT to determine whether the number of NCVs observed in promoter B that led to creation (or disruption) of a binding site for PWM A is more than expected by chance, given the total number of mutations observed in promoter B across samples within a certain cancer type. The number of promoter NCVs that create (or disrupt) a binding site for PWM A in promoter B follows a binomial distribution $P(n, p)$, where n is number of NCVs in promoter B across patients, and p is the probability that an NCV in B creates (or disrupts) a binding site for PWM A.

The probability ($p$) was estimated as:

$$p = \sum_{\substack{i=1 \\ j=1}}^{\substack{i=L \\ j=4}} F(B_i,M_j) \cdot C(PWM\,A, B_i, M_j) \quad (2)$$

where $F(B_i, M_j)$ is the probability of changing the reference base at position $i$ in promoter B to the mutated base $M_j$, $C(PWM\,A, B_i, M_j)$ is 1 if mutating $B_i$ to $M_j$ leads the creation (or disruption) of a binding site for PWM A and 0 otherwise, and $L$ is the nucleotide length of promoter B. $F(B_i, M_j)$ was calculated based on the genome-wide mutational frequencies in a cancer type, whereas $C(PWM\,A, B_i, M_j)$ was determined by calculating the motif score difference between the sequence surrounding position $i$ for the reference and alternate alleles. These motif scores were obtained by querying the altered TFBS database. We used thresholds obtained from the TFMp-value algorithm[68] to determine whether a motif score is significant, and the three different thresholds selected from the ChIP-seq allelic imbalance analysis. For a

given set of tumor samples, we calculated *P(n,p)* for each PWM-promoter pair using each of three thresholds selected independently, followed by multiple hypothesis testing correction using FDR. Of note, given that for a particular threshold NCVs are deemed significant only after comparing to the expected background using the same threshold, NCVs identified as significant using the three thresholds are overlapping but not subsets of each other. For robustness and to increase confidence in our predictions, only PWM-promoter associations that were significant with an FDR < 0.01 using all three score thresholds independently were considered in subsequent analyzes. Then, we selected the NCVs from the PCAWG samples[6] located in the promoters with significant promoter-PWM associations that were associated with differential scores of the corresponding PWM. Finally, we applied the TFA-BT to tumor samples from each of the 20 cancer types, as well as to all PCAWG samples in a pan-cancer analysis to identify predicted driver NCVs (TFA-BT NCVs). Supplementary Data 1 reports the list of TFA-BT NCVs identified, their genomic information, motif scores, frequency in different cancer types, and cancer types in which the NCVs were identified as TFA-BT NCVs. In addition, we report for each of the three thresholds used whether the TFA-BT NCVs were significantly associated with loss or gain of TF binding. All PWMs significantly associated with differential TF binding to a specific NCV for at least one threshold are reported. Given that often multiple PWMs may be associated with altered TF binding, for some PWM-NCV combinations not all three thresholds may be significant.

## Computational validation of TFA-BT NCVs

To identify functional gene sets associated with the 813 genes containing TFA-BT NCVs in their promoters, we used Metascape to obtain fold-enrichments and q-values for overlaps with GO, Reactome, and PANTHER gene sets[75]. As a comparison, functional enrichments were also determined for driver genes from IntOGen[36]. Enrichments were only computed for GO Molecular Functions, GO Biological Processes, Reactome Gene Sets, and PANTHER Pathways. The Metascape filtering parameters were set to very lenient values: the min overlap parameter was set to 3 genes, the p-value cutoff to 1, and minimum enrichment to 1. Functional genes sets with q-values > 0.05 for TFA-BT and IntOGen gene lists were removed, and the remaining gene sets were manually grouped into categories to facilitate comparisons of fold-enrichments between the TFA-BT genes and IntOGen genes. Gene ontologies were classified into supra-categories to facilitate comparisons.

We also compared enrichments of essential, fitness, and prognosis genes between TFA-BT, Cancer Gene Census[37], and IntOGen[36] genes, relative to all protein-coding genes (downloaded from the HUGO Gene Nomenclature Committee at the European Bioinformatics Institute www.genenames.org; filename gene_with_protein_product.txt). The list of genes identified as essential in all cell lines in the DepMap Achilles project was downloaded from the DepMap 21Q4 release (filename CRISPR_common_essentials.csv)[76]. The list of fitness genes was derived from the Fitness/Non-Fitness Binary Matrix (filename binaryDepScores.tsv) downloaded from the DepMap ProjectScore website[77]. Only genes designated as "fitness" genes in at least 10 cell lines were considered "fitness" genes for the enrichment analyzes. The list of prognostic genes was derived from the pathology data from the Human Protein Atlas version 21.0[35] (filename pathology.tsv). Genes with reported p-values (from Kaplan–Meier log-rank tests of the correlation between the mRNA level of each gene and survival of patients in a specific cancer type) for one or no cancer types were discarded. For the remaining gene-cancer pairs, p-values associated with favorable or unfavorable prognosis were adjusted using an FDR correction and further filtered for q-values of less than 0.01. Genes passing this threshold in at least one cancer type were considered prognostic.

Odds ratios and p-values for enrichments of essential, fitness, and prognosis genes among the TFA-BT, Cancer Gene Census, and IntOGen genes were computed using Fisher's exact tests. Enrichments of essential genes used the list of all protein-coding genes as the background, enrichments of fitness genes used the list of all genes in the unfiltered file downloaded from the ProjectScore website, and enrichments of prognostic genes used the list of all genes in the unfiltered file downloaded from the Human Protein Atlas website. Confidence intervals for the proportions of enriched genes were computed using Wald intervals.

Structural variation has been associated with changes in gene expression. We obtained genes associated with changes in gene expression caused by structural variation across 21 TCGA cohorts[38] (May 25, 2020), and considered genes with altered gene expression in more than five cancer types. We then calculated the enrichment of these genes in the 813 TFA-BT gene set using a proportional comparison test.

## Association of TFA-BT NCVs and gene expression

Aligned BAM files corresponding to 1366 samples were downloaded from ICGC. BAM files were converted to FASTQ files using the SAMtools fastq function[70]. Then, Salmon[78] was used to quantify the expression of the human transcriptome (Ensembl, May 30, 2019) in transcripts per million (TPM). The expression of each gene transcript was added to obtain the gene TPM expression. For each gene, differential expression was determined between patients with and without TFA-BT NCVs in the gene promoter. Only genes with at least five ICGC donors within the same type of cancer and at least one TFA-BT NCV in the promoter were considered. A student's *t*-test was performed to determine differential expression and *P*-values were adjusted with FDR. Expression allelic imbalance for *BCL2* in lymphoid cancer was calculated as the ratio between reads corresponding to alleles of common SNVs within transcribed regions reported by PCAWG.

## MPRA library construction

The MPRA library was constructed as previously described[39]. Briefly, oligos were synthesized (Agilent Technologies) as 230 bp sequences containing 200 bp of genomic sequences and 15 bp of adaptor sequence on either end. Unique 20 bp barcodes were added by PCR along with additional constant sequences for subsequent incorporation into a backbone vector (addgene #109035) by Gibson assembly.

The oligo library was expanded by electroporation into NEB 10-beta E. coli, and the resulting plasmid library was sequenced by Illumina 2 × 150 bp chemistry to acquire oligo-barcode pairings. The library underwent restriction digestion using AsiSI, and GFP with a minimal TATA promoter was inserted by Gibson assembly resulting in the 200 bp oligo sequence positioned directly upstream of the promoter and the 20 bp barcode residing in the 3′ UTR of GFP. After library expansion in E. coli, the final MPRA plasmid library was sequenced by Illumina 1 × 26 bp chemistry to acquire a baseline representation of each oligo-barcode pair within the library.

The library is comprised of nine sets of test sequences: 1) TFA-BT NCVs (2555 NCVs); 2) TFA-BT – unobserved (534 NCVs), corresponding to random NCVs that correspond to significant TF-promoter pairs by TFA-BT but that were not observed in the PCAWG cohort; 3) ChIP-seq allelic imbalance (281 NCVs), corresponding to a subset of NCVs that showed allelic imbalance in TF binding in ENCODE; 4) Reported driver NCVs (17 NCVs), corresponding to well characterized cancer driver NCVs; 5) Reported germline NCVs (97 NCVs), corresponding to disease-associated NCVs that show differential transcriptional activity or TF binding by enhanced yeast one-hybrid assays; 6) Random NCVs in reported genes (1298 NCVs), corresponding to random NCVs observed in PCAWG within the promoters of genes reported to have high mutational burden; 7) No predicted TF binding (500 NCVs), corresponding to NCVs that do not reside within TF binding sites; 8) No differential binding (500 NCVs), corresponding to NCVs with no predicted differential TF binding; and 9) Non-driver differential TF binding (500 NCVs), corresponding to random NCVs in non-driver

gene promoters with predicted differential TF binding (Supplementary Data 7).

In addition, we included three classes of technical controls to evaluate the sensitivity of each MPRA experiment. 1) Negative controls (506 sequences) were selected from previous experiments for not having activity across multiple cell types. 2) Positive activity controls (119 sequences) were selected for activity in multiple cell types from previous experiments. Elements were selected across the activity spectrum (e.g., not just high expressors). 3) MPRA emVar controls (64 NCVs) measuring allelic effect variants selected for showing allelic skew in multiple cell types from previous experiments (Supplementary Data 7).

## MPRA library transfection into cell lines
Jurkat cells (ATCC - TIB-152) were grown in RPMI with 10% FBS to a density of 1 million cells per mL prior to transfection. HT-29 cells (ATCC - HTB-38) were cultured in Mocoy's 5a media with 10% FBS, and SK-MEL-28 cells (ATCC - HTB-72) in EMEM supplemented with 10% FBS. Six electroporation replicates were performed on separate days by collecting 90 million cells and splitting across nine 100 uL transfections each containing 10 ug of MPRA plasmid. Cells were electroporated with the Neon Transfection System (100 µl kit) using three pulses at 1350 V for 10 ms for Jurkat cells, two pulses at 1300 V for 20 ms for HT-29 cells, and one pulse at 1200 V for 40 ms for SK-MEL-28 cells. After transfection each replicate was split between two T-175 flasks with 150 mL of culture media for recovery. After 48 h, the cells were pelleted, washed three times with PBS, and stored at −80 °C for later RNA extraction.

## RNA extraction and MPRA RNA-seq library generation
RNA for all cell lines was extracted from frozen cell pellets using the Qiagen RNeasy Maxi kit. Half of the isolated total RNA underwent DNase treatment and a mixture of three GFP-specific biotinylated primers (#120, #123 and #126)(Supplementary Table 1a) were used to capture GFP transcripts with Streptavidin C1 Dynabeads (Life Technologies). An additional DNase treatment was performed. cDNA was synthesized from GFP mRNA using SuperScript III and purified with AMPure XP beads. Quantitative PCR using primers specific for the GFP transcript (#781 and #782)(Supplementary Table 1a) was used to measure GFP transcript abundance in each sample. Replicates within each cell type were diluted to approximately the same concentration based on the qPCR results. Illumina sequencing libraries were constructed using a two-step amplification process to add sequencing adapters and indices. An initial PCR amplification with NEBNext Ultra II Q5 Master Mix and primers 781 and 782 were used to extend adapters. To minimize over-amplification during library construction, the number of PCR cycles used in the first amplification was selected based on where linear amplification began for each cell type (Jurkat: 10 cycles, SK-MEL-28 & HT-29: 13 cycles). A second 6 cycle PCR using NEBNext Ultra II Q5 Master Mix added P7 and P5 indices and flow cell adapters (Supplementary Table 1b). For SK-MEL-28 samples we failed to recover enough product during the first amplification and processed the second total RNA aliquot using the same protocol, pooling the two preparations prior to sequencing. The resulting MPRA RNA-tag libraries were sequenced using Illumina single-end 31 bp chemistry (with 8 bp index read), clustered at 80–90% maximum density on a NextSeq High Output flow cell.

## MPRA quality control and data analysis
To evaluate the quality of our MPRA libraries we determined the number of barcodes per sequence, the correlation between replicates, and the activity of positive and negative controls. The oligo library was covered by a mean of 215 barcodes in the plasmid library, 182 barcodes in Jurkat, 106 barcodes in HT-29, and 71 barcodes in SK-MEL-28 cells (Supplementary Fig. 10a) and we determined that 97.6% of oligos in the plasmid library, 96.8% in Jurkat, 94% in HT-29 and 87.9% in the SK-MEL-28 libraries were recovered with 10 or more barcodes, respectively. We observed high correlations of oligo counts in the biological replicates for Jurkat and SK-MEL-28 cells (pearson R ranged from 0.96–1) and reasonable correlations between HT-29 replicates (pearson R ranged from 0.85–0.98) (Supplementary Fig. 10b). We also show that our MPRA experiments in all three cell lines clearly distinguish the activity of positive and negative controls (Supplementary Fig. 10c–e). These three measures illustrate the high-quality of the MPRA experiments.

Data from the MPRA was analyzed as previously described[39]. Briefly, the sum of the barcode counts for each oligo were provided to DESeq2 (v.1.28.0)[79] and replicates were median normalized followed by an additional normalization of the RNA samples to center the average RNA/DNA activity distribution of the 506 negative control sequences over a log2 fold change of zero. This normalization was performed independently for each cell type. Dispersion-mean relationships were modeled for each cell type independently and used by DESeq2 in a negative binomial distribution to identify oligos showing differential expression relative to the plasmid input. Oligos passing a false discovery rate (FDR) threshold of 1% were considered to be active. For sequences that displayed significant MPRA activity, a paired t-test was applied on the log-transformed RNA/plasmid ratios for each experimental replicate to test whether the reference and alternate allele had similar activity (Supplementary Data 2). An FDR threshold of 5% was used to identify SNPs with a significant skew in MPRA activity between alleles (allelic skew).

## Mutational signatures for MPRA validated drivers
NCVs can be caused by multiple mutational processes such as UV-light. We used ICGC probabilities for each NCV-donor combination to assign them a given mutational process if its probability is greater than 0.5, as described[9]. Then, we compared the MPRA and CASCADE validation rates for TFA-BT NCVs associated with different mutational signatures. We used UV-light associated signatures[9] BI_COMPOSITE_SNV_SBS7a_S, BI_COMPOSITE_SNV_SBS7b_S, BI_COMPOSITE_SNV_SBS7c_S, BI_COMPOSITE_SNV_SBS3_P, BI_COMPOSITE_SNV_SBS55_S, BI_COMPOSITE_SNV_SBS67_S, BI_COMPOSITE_SNV_SBS75_S.

## Cell culture and nuclear extraction for CASCADE
Jurkat cells, were obtained from ATCC (TIB-152). The cells were grown in suspension in RPMI 1640 Glutamax media (Thermofisher Scientific, Catalog #72400120) with 10% heat-inactivated fetal bovine serum (Thermofisher Scientific, Catalog #132903). T175 (Thermofisher Scientific, Catalogue #132903) non-treated flasks were used when culturing Jurkat cells for experiments. Cells were grown in 50 mL of media when being cultured in T175 flasks.

SK-MEL-28 cells were obtained from the Tewhey lab to ensure the same cells used for the MPRA experiments were used for the CASCADE experiments. The cells were cultured using EMEM media (ATCC, Catalog #30-2003) with 10% heat-inactivated fetal bovine serum (Thermofisher Scientific, Catalog #132903). Cells were grown in 30 mL of media when being cultured in T225 flasks for adherent cells (Corning, Catalog #35138).

Nuclear extracts were obtained as previously described[42,80], with modifications detailed below. To harvest nuclear extracts from Jurkat cells, the cells were collected in a falcon tube and placed on ice. To harvest nuclear extracts from SK-MEL-28 cells, the media was aspirated off and the cells were washed once with 1X PBS (Thermofisher Scientific, Catalog #100010049). Once the 1X PBS used to wash the cells was aspirated off, enough 1X PBS was mixed with 0.1 mM Protease Inhibitor (Sigma-Aldrich, Catalogue #P8340) to cover the cells was added to each flask. A cell scraper was used to dislodge the cells from the flask, and cells were collected in a falcon tube and placed on ice. Jurkat and SK-MEL28 cells were pelleted by centrifugation at 500 × g for 5 min at 4 °C. Both pellets were washed with 2 mL of 1X PBS with Protease Inhibitor and pelleted again at 500 × g for 2 min at 4 °C. To lyse the plasma membrane, the cells were resuspended in Buffer A (1 mL Buffer A for

Jurkat cells, 1.5 mL Buffer A for SK-MEL28 cells) (10 mM HEPES, pH 7.9, 1.5 mM MgCl, 10 mM KCl, 0.1 mM Protease Inhibitor, Phosphatase Inhibitor (Santa-Cruz Biotechnology, Catalog #sc-45044), 0.5 mM DTT (Sigma-Aldrich, Catalog #4315) and incubated for 10 min on ice. After the 10 min incubation, Igepal detergent (final concentration of 0.1%) was added to the cell and Buffer A mixture and vortexed for 10 s. To separate the cytosolic fraction from the nuclei, the sample was centrifuged at $500 \times g$ for 5 min at 4 °C to pellet the nuclei. The cytosolic fraction was collected into a separate microcentrifuge tube. The pelleted nuclei were then resuspended in Buffer C (100 μL for Jurkat nuclei and 150 μL for SK-MEL-28 nuclei) (20 mM HEPES, pH 7.9, 25% glycerol, 1.5 mM MgCl, 0.2 mM EDTA, 0.1 mM Protease Inhibitor, Phosphatase Inhibitor, 0.5 mM DTT, and 420 mM NaCl) and then vortexed for 30 s. To extract the nuclear proteins (i.e., the nuclear extract), the nuclei were incubated in Buffer C for 1 h while mixing at 4 °C. To separate the nuclear extract from the nuclear debris, the mixture was centrifuged at $21,000 \times g$ for 20 min at 4 °C. The nuclear extract was collected in a separate microcentrifuge tube and flash frozen using liquid nitrogen. Nuclear extracts were stored at –80 °C.

## CASCADE PBM experimental methods

All experiments were performed using the 4-chambered, 4x180K Agilent microarray platform (design details described below). DNA microarrays were double stranded as described in Berger et al.[81]. PBM experiments using cell extracts were performed following the protocols previously described[80,82] and outlined below. The double-stranded microarray was pre-wetted in HBS + TX-100 (20 mM HEPES, 150 mM NaCl, 0.01% Triton X-100) for 5 min and then de-wetted in an HBS bath. Each of the microarray chambers were then incubated with 180 μL of nuclear extract binding mixture for 1 h in the dark. Nuclear extract binding mixture (per chamber): 400–600 μg of nuclear extract; 20 mM HEPES (pH 7.9); 100 mM NaCl; 1 mM DTT; 0.2 mg/mL BSA; 0.02% Triton X-100; 0.4 mg/mL salmon testes DNA (Sigma-Aldrich, Catalog #D7656). The microarray was then rinsed in an HBS bath containing 0.1% Tween-20 and subsequently de-wetted in an HBS bath. After the nuclear extract incubation, the microarray was incubated for 20 min in the dark with 20μg/mL primary antibody for the TF or COF of interest diluted in 180 μL of 2% milk in HBS (Supplementary Table 2). The following primary antibodies were used to probe cofactors on the arrays: BRD4 (ThermoFisher Scientific, Cat # A301-985A50); TBL1XR1 (Santa Cruz, Cat # sc100908); SRC1 (Santa Cruz, Cat # sc32789x); MOF (ThermoFisher Scientific, Cat # A300-992A); RBBP5 (ThermoFisher Scientific, Cat # A300-109A); NCOR1 (ThermoFisher Scientific, Cat # A301-145A). These antibodies were validated by western blot. After the primary antibody incubation, the array was first rinsed in an HBS bath containing 0.1% Tween-20 and then de-wetted in an HBS bath. Microarrays were then incubated for 20 min in the dark with 10 μg/mL of either Alexa488- or Alexa647-conjugated secondary antibody (see Supplementary Table 2) diluted in 180 μL of 2% milk in HBS. The following fluorescently labeled secondary antibodies were used in our CASCADE experiments, and were species matched to the primary antibodies described above: Goat anti-rabbit IgG (H + L) Highly Cross-absorbed Secondary Antibody, Alexa Fluor 647 (ThermoFisher, Cat #A32733); Goat anti-mouse IgG (H + L) Highly Cross-absorbed Secondary Antibody, Alexa Fluor 488 (ThermoFisher, Cat # A11029). Excess antibody was removed by washing the array twice for 3 min in 0.05% Tween-20 in HBS and once for 2 min in HBS in Coplin jars as described above. After the washes, the microarray was de-wetted in an HBS bath. Microarrays were scanned with a GenePix 4400 A scanner and fluorescence was quantified using GenePix Pro 7.2. Exported fluorescence data were normalized with MicroArray LINEar Regression[83].

## CASCADE microarray designs

CASCADE experiments were performed using custom-designed microarrays (Agilent Technologies Inc, AMADID 086310 and 086772, 4x180K format). Microarray probes are all 60 nucleotides (nt) long and of the format: "GCCTAG" 5-prime flank sequence−- 26-nt variable sequence−- "CTAG" 3-prime flank sequence−- "GTCTTGATTCGCTT-GACGCTGCTG" 24-nt common primer (Supplementary Data 8). For each unique probe sequence (i.e., unique 26-nt variable region) five replicate probes are included on the microarray with the variable sequence in each orientation with respect to the glass slide (i.e., 10 probes total per unique variable sequence).

**Design 1 (Agilent AMADID 086310): Microarray design for profiling Ref/Alt impact.** This microarray was designed to profile the impact of NCVs on COF binding by comparing the binding to reference (Ref) and alternate (Alt) probes. The design included 2956 Ref/Alt paired probe sets that include: 2555 TFA-BT NCVs, 17 literature-reported driver NCVs, and 384 background NCVs (Supplementary Data 8). The background NCVs were selected from those NCVs for which the TFA-BT algorithm found no predicted binding of any TF. A priori we do not know where within a TF binding site a NCV will reside, so probe sequences were designed such that each NCV was represented in three separate DNA registers in our microarray (i.e., NCV centered in each DNA probe, or offset by 5 nt in either direction, Supplementary Fig. 3a, b). Using this design, each Ref/Alt pair (i.e., each NCV assayed) required 60 individual probes on our array (3 registers x 10 replicates x 2 Ref/Alt-variants).

**Design 2 (Agilent AMADID 086310): Microarray design for determining COF motifs.** This microarray was designed to determine COF recruitment motifs for each NCV loci. The design is based on the exhaustive mutagenesis approach outlined in Bray & Hook et al.[42]. where all possible single-nucleotide variant (SV) probes of a defined genomic locus are included as probes in the microarray. By profiling the differential binding of a COF to all SV probes we can directly determine a motif/logo for that COF and genomic loci as described in Bray & Hook et al. (details below). The design included probes to evaluate motifs at 359 NCVs identified as significant by both CASCADE (differential COF recruitment using Design 1 microarray) and MPRAs (differential gene expression) (Supplementary Data 9). In our initial NCV screen using the Design 1 microarray, for each NCV we evaluated the differential COF binding to probes in the three different NCV registers (i.e., NCV centered or offset, see above) and two orientations with respect to the glass slide. For the Design 2 microarray, we selected the probe register and orientation that gave the largest differential COF binding in our initial NCV screen, and used this 'best register' probe (hereafter referred to as the 'seed' sequence) along with all SV probes covering the 26-nt genomic locus. Furthermore, for the starting seed sequence we used either the Ref or the Alt probe based on which had the strongest COF binding in our initial screen. We note that this specific choice of Ref or Alt as the starting seed probe was generally consistent across all different COF experiments. Each unique 26-nt sequence was represented by 5 replicate probes. Using this design, each NCV loci was characterized using 395 individual probes on our microarray: (1 seed + 3 variants x 26 positions) x 5 replicates.

## CASCADE computational analysis

Image analysis and spatial detrending of the microarray fluorescence intensities were performed as previously described[81,83]. Probe fluorescence values were transformed to a z-score (as previously described[80]) using the fluorescence distribution of a set of background probes included on each microarray.

**Design 1: Microarray design for profiling Ref/Alt impact.** To determine differential COF binding due to each NCV, probe intensities were compared between the Ref and Alt probes. For each NCV, differential binding was assessed independently to all six sequences representing that NCV (i.e., three NCV registers and two orientations). For each of the six sequences, the significance of the differential binding was

assessed using a two-sided Student's *t*-test between the 5 replicate probes for the Ref and Alt alleles. Finally, an aggregate, multiple hypothesis-corrected *p*-value for differential binding was determined using Fisher's method (sum log p-values) and the six independent *p*-values. The magnitude of the differential binding was quantified using a " Δz-score" computed as the difference in the mean z-score for the Ref probes (all registers, orientations, and replicates) and the Alt probes. Therefore, for each NCV we assessed the magnitude (Δz-score) and significance (aggregate p-value) of the differential COF binding. We annotated NCVs as differentially bound in each experiment if they met the following criteria: (1) the z-score of Ref or Alt allele > 2.0; (2) delta z-score > 2.0; (3) aggregate p-value $<10^{-3}$. NCVs were called differentially bound if they met the above criteria in both replicate CASCADE experiments.

**Design 2 (Agilent AMADID 086310): Microarray design for determining COF motifs.** COF motifs were determined by evaluating the z-scores for the seed and SV probes representing each NCV as previously described[42,80]. COF motifs can either be represented as a Δz-score matrix, which is akin to an energy matrix that evaluates the change in binding magnitude for each nucleotide variant, or as a position probability matrix (PPM) that is based on a probabilistic model relating base frequencies and binding energies[84]. We use Δz-score matrices to directly show of the impact of base identify on binding and use PPMs to compare against motifs in public databases which almost exclusively represent motifs as PPMs. Δz-score matrices for a locus are determined using z-scores from the seed probe ($z_{seed}$) and three SV probes at each of the 26 base positions across the locus. The Δz-score matrix values are based on the z-score differences from the median, calculated independently for each position (i) along the probe:

$$\triangle z_{i,j} = z_{i,j} - median_{j=A,C,G,T}\left(z_{i,j}\right) \qquad (3)$$

where *i* indicates the nucleotide position (1 to 26) and *j* indicates the nucleotide (A,C,G,T). The median at position *i* is determined over the seed sequence and three probes with variant nucleotide at position *i*. PPMs are determined by transforming the same z-scores in a different manner:

$$PPM_{i,j} = \frac{\exp(\beta^* z_{i,j})}{\sum_j \exp(\beta^* z_{i,j})} \qquad (4)$$

where *i* indicates the nucleotide position (1 to 26), *j* indicates the nucleotide (A,C,G,T), and β is an empirically determined scaling parameter:

$$\beta = 4 \quad z_{seed} < 0$$

$$\beta = 4 - \frac{z_{seed}}{2} \quad 0 \le z_{seed} \le 6$$

$$\beta = 1 \quad 6 < z_{seed}$$

PPMs for each locus were compared against PPMs from JASPAR[85] using the TomTom[86] algorithm (dist = Euclidean Distance; min_overlap = 6) using the "meme" package[87] implemented in R.

**Statistical analyzes**
Statistical analyzes and figures were generated using GraphPad Prism (v.9.2) and R stats (v.4.2).

**Reporting summary**
Further information on research design is available in the Nature Portfolio Reporting Summary linked to this article.

## Data availability

The MPRA and CASCADE data generated in this study have been deposited in the Gene Expression Omnibus database under accession code GSE218478 and GSE222436, respectively. The PCAWG data and motif analyzes used in this study are provided in the Supplementary Information/Source Data files. The cancer mutation data used in this study was obtained from PCAWG (https://dcc.icgc.org/pcawg). The TF motifs used in this study were obtained from CIS-BP (http://cisbp.ccbr. utoronto.ca) and JASPAR (https://jaspar.genereg.net). The list of prognostic genes was derived from the pathology data from the Human Protein Atlas version 21.0 (https://www.proteinatlas.org/about/download). The ChIP-seq data used for the allelic imbalance analysis was obtained from the ENCODE Project (https://www.encodeproject.org and Supplementary Data 5). Additional information required to reanalyze the data reported in this paper is available from the lead contacts upon request. Source data are provided with this paper.

## Code availability

Original code for the TFA-BT has been deposited on GitHub (https://github.com/fuxmanlab/noncoding_drivers and www.github.com/fuxmanlab/altered_TFBS) and is publicly available. This code is also deposited in Zenodo (https://doi.org/10.5281/zenodo.7570531)[88]. Original code for the CASCADE analysis has been deposited on GitHub (https://github.com/Siggers-Lab/Carrasco-Pro-Hook-et-al.-PBM-Analysis.git) and is publicly available. Code to analyze the MPRA data is available in GitHub: MPRAmatch, MPRAcount (https://github.com/tewhey-lab/MPRA_oligo_barcode_pipeline) and MPRAmodel (https://github.com/tewhey-lab/MPRAmodel).

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

## Acknowledgements

We thank Katia Bulekova and Brian Gregor for computational and I&T assistance. We also thank Drs. Zeba Wunderlich and Ana Fiszbein for critically reading and commenting on the manuscript. This work was funded by the National Institutes of Health (NIH) grants R35 GM128625 awarded to J.I.F.B, R21 HG011289 and R01 AI51051 awarded to T.S., and R00HG008179 and R35HG011329 awarded to R.T.

## Author contributions

S.C.P and J.I.F.B. conceived the project. S.C.P., A.T.L., and J.I.F.B. developed the TFA-BT. S.C.P., D.M., D. Bray, D. Berenzy, H.H., R.T., T.S., and J.I.F.B. performed data analyzes and generated the figures. D. Berenzy and M.Y. performed the MPRA experiments. H.H. performed the CASCADE experiments. S.C.P., J.I.F.B., H.H., and T.S. wrote the manuscript. All authors read, edited, and approved the manuscript.

## Competing interests

The authors declare no competing interests.
