## [Peer review file · Nature Communications]

REVIEWER COMMENTS

Reviewer #1 (Remarks to the Author): Expert in cancer drivers, genomics, and bioinformatics

Finding driver non-coding variants (NCVs) in the cancer genome is challenging, particularly in tumour types (like skin melanoma) where the somatic mutation rate is very high at the promoter regions. In this manuscript, the authors proposed a novel computational approach, the transcription factor (TF)-aware burden test (TFA-BT), to predict driver NCVs based on their effect on the gain/loss of TF binding in the promoter regions. The authors applied this approach to PCAWG tumours and predicted 2,555 driver NCVs in the promoters of 813 genes. Further functional analyses revealed that many of these NCVs can potentially affect gene expression and this can be attributed to the perturbations in the TF or TF-cofactor binding to those mutated sites, which is novel and interesting. Overall, the manuscript was well written, and the computational and experimental findings support the conclusion convincingly.

Major comments:

1) Are these 2,555 driver NCVs predicted in 813 genes based on the $FDR < 0.01$ satisfying all three score thresholds of TFA-BT or union of the significant hits identified under each of the three score thresholds separately? The details of which are missing in the result section. Also, in Supplementary Table 1, there are rows (driver NCVs) with zero value for one of the following columns 'Loss (case 1)', 'Loss (case 2)', 'Loss (case 3)', for example. Does that mean those NCVs were identified through one of three score thresholds? This needs to be clarified as well.

2) L515-519: Regarding the three score thresholds/parameters of TFA-BT: “2) at least one allele passed the motif binding threshold and the difference in score between alleles was greater than two, and 3) at least one allele passed the motif binding threshold and the difference in score between alleles was greater than three.”

The rationale behind the parameter settings of (2) and (3) is unclear, because parameter (2) seems to be inclusive of (3). Is this meant for a different level of stringency? However, when all three parameters have been considered to find the significant and robust hits (as mentioned on Page 24 L551-554), these parameters (2 and 3) seem redundant.

3) L543-544: “ $F(B_i, M_j)$ was calculated based on the genome-wide mutational frequencies in a cancer type”

Are these mutational frequencies calculated based on mutation type alone or considering their immediate 5' and 3' sequence context (e.g., tri-nucleotide context) as well? The sequence context might help to model the mutation rate better, especially following the mutational processes/signatures observed in the tumour types. See related comment #7 below.

4) L107-109: "Most TFA-BT NCVs (65%) were obtained from skin cancer (Fig. 1b). This is not only related to skin cancer samples having the largest number of SNVs, but also to a higher fraction of these being predicted as TFA-BT NCVs (Supplementary Fig. 1a)."

This could be partly explained by the higher number of somatic mutations at the TF binding sites in the promoter regions in skin melanoma (<https://doi.org/10.1038/nature17437> and <https://doi.org/10.1038/nature17661>). Perhaps, in Supplementary Fig. 1a, instead of "All SNVs" the authors may consider plotting the number of mutations in the promoter regions so that one can see what fraction of the mutations in the promoter regions are predicted as driver NCVs.

5) L135-138: Did any of the novel driver NCVs reported in RPL13A, TEDC2 and PES1 show changes in gene expression (MPRAs) or TF-cofactor binding (CASCADE) analyses?

6) Have the authors checked whether the driver NCVs predicted with TFA-BT NCVs correlate with gene expression (or chromatin accessibility) changes in the patient samples (if any available)? Alternatively, if the whole genome sequence of the cancer cell lines (Jurkat, SK-MEL-28, HT-29) are available, the authors may check for the presence of predicted driver NCVs (from the patient tumour) in the cell line genome and correlate the effect of those mutations with the gene regulation (accessibility/TF binding) and expression in the respective cell lines.

7) Given that the core motif (GGAA) of ETS consensus sequence is enriched with di-pyrimidines, which have a higher probability to mutate with respect to UV-light associated mutational signatures, is the finding that widespread ETS motif perturbation is expected by chance (at least in melanoma), as compared to other TFs?

8) Why only EGR1 and RNF20 gene promoters were chosen to present in Figure 5, as there are other genes with similar effects reported in Supplementary Figure 8?

9) In Figure 5, some of the NCVs in EGR1 and RNF20 promoters are adjacent to each other. Given that the double mutations (eg., CC>TT) or MNVs are prominent in melanoma. Are these reported mutations coming from the same samples? If so, are these mutations tested together (instead of independent) in the functional assays?

Minor comment:

1) Page 5, Line 92: “predicted the alter the binding” -> “predicted to alter the binding”

Reviewer #3 (Remarks to the Author): Expert in cancer proteomics and protein-binding microarrays

In the manuscript by Carrasco & Hook et al. entitled “Widespread perturbation of ETS factor binding sites in cancer”, the authors perform a relevant study that enables the prediction of driver non-coding variants using a novel transcription factor aware burden test based on a model of coherent TF function in promoters.

The authors applied the test to the Pan-Cancer Analysis of Whole Genomes cohort predicting 2555 driver NCVs in the promoters of 813 genes in 20 different cancers. The authors then validate partially the results using three different cell lines from melanoma, colon cancer and lymphoma determining that 765 candidate driver NCVs alters the transcription activity and 510 to differential binding of TF-cofactor regulatory complexes.

Results sound really interesting. The manuscript is well written.

Although it would be recommended to validate the results using more than one cell line per cancer type, it is believed that the manuscript is good enough as for the journal.

Minor points/typos:

Introduction: line 60 BLC6 should read BCL6. Please rewrite also the sentence; some words are missing to give the sentence the exact meaning.

Reviewer #4 (Remarks to the Author): Expert in MPRAs, functional genomics, and gene regulation

In the manuscript entitled “Widespread perturbation of ETS factor binding sites in cancer”, Pro and colleagues developed a novel method called TF-aware burden test (TFA-BT) that allows to predict cancer-driver non-coding variants (NCVs) in the disease gene promoters. This was developed based on the assumption that the driver NCVs are associated with the change of TF binding motifs (creating or disrupting) at different positions in a promoter. Therefore, the TFA-BT allows the analysis of disease risks with high resolution (even 1nt), as well as understanding the molecular mechanism of the disease, such as TF binding, cofactor recruitment, etc. In fact, they found 2,555 potential cancer-driver NCVs, which is more efficient than similar approaches. Furthermore, they experimentally validated the results using cutting edge technologies, MPRA and CASCADE. These analyses showed that the cancer driver NCVs are widespread and associated with ETS factor binding sequences. Their approach and results can be impactful not only in the cancer research field but broadly as a landmark. I would recommend this manuscript to be published in Nature Communications.

1. Because their analysis only focuses on gene promoters (from -2,000 to +250bp of the TSS) in this manuscript, their finding, especially the enrichment of cancer driver NCVs at ETS binding sites, could have been biased to promoter regions, thus may not be applicable genomewide, as the authors already mentioned “The presence of composite ETS sites is consistent with their tendency to cluster in human promoters” (Page 14, line 303). I would like to suggest the authors to clearly state the above caveat in the manuscript to avoid misleading. Or, I was wondering if there is evidence or prediction that the enrichment of NCVs in ETS binding sites can be observed in distal regions in cancer genome.

2. MPRA method and results were not sufficiently provided. Please visualize the following information in main or supplementary figures.

i. What are positive controls (as shown in Figure 2b) and negative control sequences used in the MPRA.

ii. As the barcodes are random, how many (on average) barcodes were associated per sequence?

iii. Related to the statements, “only a subset of DNA regions are active (page 8, line 169)”, “Oligos passing a false discovery rate (FDR) threshold of 1% were considered to be active (page 28, line 658)”, please provide the actual number of active sequences and those with emVers were detected at a certain threshold.

iv. Reproducibility of the MPRA experiments was not clear. Please show more detail about the reproducibility of barcode expression, activity of each sequence, and emVers.

Reviewer #1 (Remarks to the Author): Expert in cancer drivers, genomics, and bioinformatics

Finding driver non-coding variants (NCVs) in the cancer genome is challenging, particularly in tumour types (like skin melanoma) where the somatic mutation rate is very high at the promoter regions. In this manuscript, the authors proposed a novel computational approach, the transcription factor (TF)-aware burden test (TFA-BT), to predict driver NCVs based on their effect on the gain/loss of TF binding in the promoter regions. The authors applied this approach to PCAWG tumours and predicted 2,555 driver NCVs in the promoters of 813 genes. Further functional analyses revealed that many of these NCVs can potentially affect gene expression and this can be attributed to the perturbations in the TF or TF-cofactor binding to those mutated sites, which is novel and interesting. Overall, the manuscript was well written, and the computational and experimental findings support the conclusion convincingly.

We thank the reviewer for the positive comments and for the thoughtful suggestions that helped further improve our manuscript. Below are our point-by-point responses:

Major comments:

1) Are these 2,555 driver NCVs predicted in 813 genes based on the $FDR < 0.01$ satisfying all three score thresholds of TFA-BT or union of the significant hits identified under each of the three score thresholds separately? The details of which are missing in the result section. Also, in Supplementary Table 1, there are rows (driver NCVs) with zero value for one of the following columns 'Loss (case 1)', 'Loss (case 2)', 'Loss (case 3)', for example. Does that mean those NCVs were identified through one of three score thresholds? This needs to be clarified as well.

We thank the reviewer for pointing this out. We used the three thresholds independently to predict TFA-BT NCVs and then selected the 2,555 NCVs that were identified using all three thresholds. Although this is explained in the Methods section, we agree with the reviewer that adding an explanation in the Results section would be beneficial. We have now added this in line 98-101 of the revised manuscript: *“Three different threshold-based approaches were used to predict differential TF binding to NCV alleles and for the subsequent steps in the TF-ABT. For robustness we only considered as TFA-BT NCVs those that were deemed significant ($FDR < 0.01$) using all three thresholds (see Methods).”*

Regarding Supplementary Table 1, an NCV was annotated as a TF-ABT NCV if it was identified using all three thresholds with at least one PWM. Once an NCV was selected, we reported all PWMs that were deemed significant for any of the three thresholds. To clarify this, we have added an explanation in the Methods section of the revised manuscript in lines 515-521: *“Supplementary Table 1 reports the list of TFA-BT NCVs identified, their genomic information, motif scores, frequency in different cancer types, and cancer types in which the NCVs were identified as TFA-BT NCVs. In addition, we report for each of the three thresholds used whether the TFA-BT NCVs were significantly associated with loss or gain of TF binding. All PWMs significantly associated with differential TF binding to a specific NCV for at least one threshold are reported. Given that often multiple PWMs may be associated with altered TF binding, for some PWM-NCV combinations not all three thresholds may be significant.”*

2) L515-519: Regarding the three score thresholds/parameters of TFA-BT: “2) at least one allele passed the motif binding threshold and the difference in score between alleles was greater than two, and 3) at least one allele passed the motif binding threshold and the difference in score between alleles was greater than three.”

The rationale behind the parameter settings of (2) and (3) is unclear, because parameter (2) seems to be inclusive of (3). Is this meant for a different level of stringency? However, when all three parameters have been considered to find the significant and robust hits (as mentioned on Page 24 L551-554), these parameters (2 and 3) seem redundant.

We thank the reviewer for the comment. Although, in principle, it would appear that a threshold of 2 is inclusive of a threshold of 3, this is not always the case. The reason is that sometimes, using a threshold of 3 identifies a PWM as having more NCV instances within a promoter than expected by chance (also considering the chance of a random variant in the promoter disrupting the PWM with a threshold of 3), but when the threshold is changed to 2, this is no longer significant (for example, because there are many possible random variants in the promoter with a threshold between 2 and 3). We have now explained this situation in the methods section in lines 505-508: *“Of note, given that for a particular threshold NCVs are deemed significant only after comparing to the expected background using the same threshold, NCVs identified as significant using the three thresholds are overlapping but not subsets of each other.”*

3) L543-544: “F(Bi, Mj) was calculated based on the genome-wide mutational frequencies in a cancer type”

Are these mutational frequencies calculated based on mutation type alone or considering their immediate 5' and 3' sequence context (e.g., tri-nucleotide context) as well? The sequence context might help to model the mutation rate better, especially following the mutational processes/signatures observed in the tumour types. See related comment #7 below.

We thank the reviewer for pointing this out. In this instance of TFA-BT, we used the mutational frequency without considering the tri-nucleotide context, which still proved effective at identifying functional NCVs by MPRAs and CASCADE. Future iterations of the approach will consider the tri-nucleotide context as we agree with the reviewer that it may further improve predictions.

4) L107-109: “Most TFA-BT NCVs (65%) were obtained from skin cancer (Fig. 1b). This is not only related to skin cancer samples having the largest number of SNVs, but also to a higher fraction of these being predicted as TFA-BT NCVs (Supplementary Fig. 1a).”

This could be partly explained by the higher number of somatic mutations at the TF binding sites in the promoter regions in skin melanoma (<https://doi.org/10.1038/nature17437> and <https://doi.org/10.1038/nature17661>). Perhaps, in Supplementary Fig. 1a, instead of “All SNVs” the authors may consider plotting the number of mutations in the promoter regions so that one can see what fraction of the mutations in the promoter regions are predicted as driver NCVs.

We thank the reviewer for this suggestion. We have revised Supplementary Fig. 1a to consider only promoter NCVs as background and we observe a similar trend than when considering all SNVs. We have also modified the results section lines 111-113 to clarify this: *“This is not only related to skin cancer samples having the largest number of promoter NCVs, but also to a higher fraction of these being predicted as TFA-BT NCVs (Supplementary Fig. 1a).”*

5) L135-138: Did any of the novel driver NCVs reported in RPL13A, TEDC2 and PES1 show changes in gene expression (MPRAs) or TF-cofactor binding (CASCADE) analyses?

We thank the reviewer for the suggestion. We did observe some TFA-BT NCVs for the above genes that were validated by MPRAs and/or CASCADE:

TEDC2: 2 NCVs validated by MPRAs, one of which shows differential binding of some of the cofactors tested.

PES1: 1 NCV validated by MPRAs, but did not show differential binding of any of the 6 cofactors tested.

RPL13A: no NCVs validated by MPRAs in the cell lines tested.

However, we consider the cases we used to illustrate our integrated approach (EGR1, RNF20, FBXL18, and PARS2) to be better examples also allowing us to show mechanistic similarities and differences between NCVs within a promoter.

6) Have the authors checked whether the driver NCVs predicted with TFA-BT NCVs correlate with gene expression (or chromatin accessibility) changes in the patient samples (if any available)? Alternatively, if the whole genome sequence of the cancer cell lines (Jurkat, SK-MEL-28, HT-29) are available, the authors may check for the presence of predicted driver NCVs (from the patient tumour) in the cell line genome and correlate the effect of those mutations with the gene regulation (accessibility/TF binding) and expression in the respective cell lines.

We thank the reviewer for this excellent suggestion. We have now included an additional analyses of gene expression in patient tumor samples (lines 195-212 in the Results section and lines 564-575 in the Methods section) where we associate TFA-BT NCVs with tumor gene expression and allelic expression. We found several examples in which TFA-BT NCVs are associated with differential expression across tumor samples or allelic expression within samples, supporting our conclusions. However, we note that this analysis is complicated by the difficulty in comparing gene expression levels across biological samples, the small number of samples with available TFA-BT NCVs per gene, and that the stage in cancer development when altered expression may occur is unknown. We also looked for evidence of allele specific binding by examining thousands of available ChIP-seq datasets from ENCODE using cell lines but were unable to find examples of the TFA-BT variants in these datasets.

7) Given that the core motif (GGAA) of ETS consensus sequence is enriched with di-pyrimidines, which have a higher probability to mutate with respect to UV-light associated mutational signatures, is the finding that widespread ETS motif perturbation is expected by chance (at least in melanoma), as compared to other TFs?

This is an interesting point, and one that we had considered when putting together our manuscript. We agree with the reviewer that based on the GG dinucleotide we might expect increased UV light-mediated mutations in ETS sites. This bias has also been previously noted [Refs 48,49]. However, these previous studies did not attempt to show whether these mutations were functional. Here, we show that these mutations are functional both by MPRA and CASCADE. Therefore, whether mutant ETS sites are occurring due to increase mutation rates alone, or increased mutation and subsequent selection, here we demonstrate that they are functional and are likely to play an important role in gene dysregulation in cancer. This is considered in the Discussion section, lines 396-404: *"A large fraction of ETS binding disruptions are associated with UV-light mutational signatures and are concentrated primarily in the GG doublet of the canonical 5'-GGAA-3' ETS box and downstream bases, as has been reported 48,49. Mutations at these positions have been associated with increased mutational rates at sites of ETS factor binding and potential reduced DNA repair 65,66, but are mostly considered non-functional and are, therefore, excluded from most burden tests. Here, we show that these frequent ETS-disrupting mutations have the largest transcriptional effects and disruption of COF*

binding. This suggests that excluding these mutations, as well as those associated with other mutational signatures such as APOBEC and POLE, may not be warranted.”

8) Why only EGR1 and RNF20 gene promoters were chosen to present in Figure 5, as there are other genes with similar effects reported in Supplementary Figure 8?

We have focused on EGR1 and RNF20 in Figure 5 as for these genes we had multiple NCVs with available MPRA data and CASCADE data for the six cofactors tested. There were several other cases that could have been used as examples (e.g., PARS2, FBXL18 that we included in Supplementary Fig. 8). The examples shown were selected based on: 1) the availability of both CASCADE and MPRA data (in MPRA active regions), 2) the number of mutations with available data, and 3) the known role of the genes in cancer. This allowed us to have a more in-depth discussion of the role of the expression of these genes in cancer and also to compare mechanisms between NCVs within the same promoter.

9) In Figure 5, some of the NCVs in EGR1 and RNF20 promoters are adjacent to each other. Given that the double mutations (eg., CC>TT) or MNVs are prominent in melanoma. Are these reported mutations coming from the same samples? If so, are these mutations tested together (instead of independent) in the functional assays?

Great point. We have only considered single nucleotide mutations and excluded dinucleotide mutations and indels from all the prediction analyses. We have clarified this in the methods section of the revised manuscript in lines 476-477: *“We considered only single-nucleotide variants (SNVs) and excluded multi-nucleotide variants from the analyses.”*

Minor comment:

1) Page 5, Line 92: “predicted the alter the binding” -> “predicted to alter the binding”

We have fixed the typo.

Reviewer #3 (Remarks to the Author): Expert in cancer proteomics and protein-binding microarrays

In the manuscript by Carrasco & Hook et al. entitled “Widespread perturbation of ETS factor binding sites in cancer”, the authors perform a relevant study that enables the prediction of driver non-coding variants using a novel transcription factor aware burden test based on a model of coherent TF function in promoters.

The authors applied the test to the Pan-Cancer Analysis of Whole Genomes cohort predicting 2555 driver NCVs in the promoters of 813 genes in 20 different cancers. The authors then validate partially the results using three different cell lines from melanoma, colon cancer and lymphoma determining that 765 candidate driver NCVs alters the transcription activity and 510 to differential binding of TF-cofactor regulatory complexes.

Results sound really interesting. The manuscript is well written.

Although it would be recommended to validate the results using more than one cell line per cancer type, it is believed that the manuscript is good enough as for the journal.

We thank the reviewer for the positive and encouraging comments and for the suggestion. Given the complexity of experiments, validating in multiple cell lines per cancer type would be challenging. Nevertheless, we and others have noticed that there is a marked overlap between NCVs validated by MPRA in different cell lines (Supplementary Fig. 2a). For these reasons, we avoided making claims about cell line specificity of NCVs which would have needed a much broader panel of cell lines. We anticipate a follow up study to tackle this specific question.

Minor points/typos:

Introduction: line 60 BLC6 should read BCL6. Please rewrite also the sentence; some words are missing to give the sentence the exact meaning.

We have fixed the typo and rewrote the sentence for clarity.

Reviewer #4 (Remarks to the Author): Expert in MPRA, functional genomics, and gene regulation

In the manuscript entitled “Widespread perturbation of ETS factor binding sites in cancer”, Pro and colleagues developed a novel method called TF-aware burden test (TFA-BT) that allows to predict cancer-driver non-coding variants (NCVs) in the disease gene promoters. This was developed based on the assumption that the driver NCVs are associated with the change of TF binding motifs (creating or disrupting) at different positions in a promoter. Therefore, the TFA-BT allows the analysis of disease risks with high resolution (even 1nt), as well as understanding the molecular mechanism of the disease, such as TF binding, cofactor recruitment, etc. In fact, they found 2,555 potential cancer-driver NCVs, which is more efficient than similar approaches. Furthermore, they experimentally validated the results using cutting edge technologies, MPRA and CASCADE. These analyses showed that the cancer driver NCVs are widespread and associated with ETS factor binding sequences. Their approach and results can be impactful not only in the cancer research field but broadly as a landmark. I would recommend this manuscript to be published in Nature Communications.

We thank the reviewer for the supportive and encouraging comments.

1. Because their analysis only focuses on gene promoters (from -2,000 to +250bp of the TSS) in this manuscript, their finding, especially the enrichment of cancer driver NCVs at ETS binding sites, could have been biased to promoter regions, thus may not be applicable genomewide, as the authors already mentioned “The presence of composite ETS sites is consistent with their tendency to cluster in human promoters” (Page 14, line 303). I would like to suggest the authors to clearly state the above caveat in the manuscript to avoid misleading. Or, I was wondering if there is evidence or prediction that the enrichment of NCVs in ETS binding sites can be observed in distal regions in cancer genome.

We thank the reviewer for the comment. We have clarified this in the Discussion in line 393-396: *“Our studies further identified the disruption of ETS binding sites in gene promoters as a widespread cancer mechanism. Whether disruption of ETS factor binding sites is also frequent in enhancers, which often bind a different TF repertoire than promoters, remains to be determined.”*

2. MPRA method and results were not sufficiently provided. Please visualize the following information in main or supplementary figures.

i. What are positive controls (as shown in Figure 2b) and negative control sequences used in the MPRA.

We included three classes of technical controls to evaluate the sensitivity of each MPRA experiment. (i) Negative controls were selected from previous experiments for not having activity across multiple cell types. We included 506 negative controls in the library. (ii) Positive activity controls were selected for activity in multiple cell types from previous experiments. Elements were selected across the activity spectrum (e.g., not just high expressors). We included 119 positive controls in the library. (iii) 64 NCVs measuring allelic effect variants were selected for showing allelic skew in multiple cell types from previous experiments. We have added a brief description of each of the NCV test sets and the three technical controls, as well as the numbers of sequences tested in each case, in the Methods section of the revised manuscript (lines 590-610). We have also added new Supplemental Figs, 10c-e showing the activity profile of the entire reporter library with negative and positive activity controls highlighted.

ii. As the barcodes are random, how many (on average) barcodes were associated per sequence?

We have now provided a cumulative distribution plot of barcode coverage for each oligo across cell types. The oligo library was covered by a mean of 215 barcodes in the plasmid library, 182 barcodes in Jurkat, 106 barcodes in HT-29, and 71 barcodes in SK-MEL-28 cells. We have previously observed that the quality of the activity estimates decreases substantially when fewer than 5-10 barcodes per oligo are recovered. Using 10 barcodes as a cutoff, we recovered 97.6% of oligos in the plasmid library, 96.8% in Jurkat, 94% in HT-29 and 87.9% in the SK-MEL-28 libraries, respectively.

We have added this information in the Methods section of the revised manuscript (lines 649-652): “*The oligo library was covered by a mean of 215 barcodes in the plasmid library, 182 barcodes in Jurkat, 106 barcodes in HT-29, and 71 barcodes in SK-MEL-28 cells (Supplementary Fig. 10a) and we determined that 97.6% of oligos in the plasmid library, 96.8% in Jurkat, 94% in HT-29 and 87.9% in the SK-MEL-28 libraries were recovered with 10 or more barcodes, respectively.*”

iii. Related to the statements, “only a subset of DNA regions are active (page 8, line 169)”, “Oligos passing a false discovery rate (FDR) threshold of 1% were considered to be active (page 28, line 658)”, please provide the actual number of active sequences and those with emVars were detected at a certain threshold.

Of the 2,555 TFA-BT NCVs, 1,378, 1,118, and 1,144 were in active regions in Jurkat, SK-MEL-28, and HT-29 cells, respectively. We have added this information in the Results section of the revised manuscript (lines 148-151): “*Since only a subset of DNA regions are active (show MPRA activity for either allele – 1,378 for Jurkat, 1,118 for SK-MEL-28, and 1,144 for HT-29 cells), we calculated the validation rate as the ratio of emVars over the total number of active DNA regions for each NCV category.*” The number of active regions for each of the remaining test set categories can be obtained from Supplementary Table 2.

iv. Reproducibility of the MPRA experiments was not clear. Please show more detail about the reproducibility of barcode expression, activity of each sequence, and emVars.

We observed excellent correlations of oligo counts in the biological replicates for Jurkat and SK-MEL-28 cells (pearson R ranged from 0.96-1) and reasonable correlations between HT-29 replicates (pearson R: 0.85-0.98). To provide the reader with more information regarding the quality and reproducibility of the MPRA experiments we have included 5 new supplementary

figures. This includes: (i) Supplementary Fig. 10a showing the cumulative distribution of the oligo library in each cell type covered by a minimum number of unique barcodes. (ii) Supplementary Fig. 10b is a correlation matrix for the first and last replicate of each cell line and the plasmid library showing the reproducibility between biological replicates. (iii) Supplementary Fig. 10c-e show the MPRA activity plotted against mean plasmid count with positive and negative controls highlighted. We believe these 5 plots convey the quality of the MPRA experiments. However, we are happy to provide additional plots the reviewers believe may be helpful for the readers. We have added the description of the quality analyses to the Methods section of the revised manuscript in lines 646-658.

REVIEWERS' COMMENTS

Reviewer #1 (Remarks to the Author):

The authors have adequately addressed all my comments.

Reviewer #3 (Remarks to the Author):

The authors have properly addressed reviewer's comments.

Reviewer #4 (Remarks to the Author):

The authors have addressed all my comments.